# Sea lamprey nests promote the diversity of benthic macroinvertebrate assemblages

**Marius Dhamelincourt**[1,2]*, **Jacques Rives**[1,2], **Marie Pons**[3], **Aitor Larrañaga**[4], **Cédric Tentelier**[1,2], **Arturo Elosegi**[4]

**1** Université de Pau et des Pays de l'Adour, E2S UPPA, INRAE, ECOBIOP, Saint-Pée-sur-Nivelle, France, **2** Pôle Gestion des Migrateurs Amphihalins dans leur Environnement, OFB, INRAE, Agrocampus Ouest, Univ Pau & Pays Adour/ E2S UPPA, Pau, France, **3** Bureau d'études Aquabio, Route de Créon, Saint-Germain-du-Puch, France, **4** Department of Plant Biology and Ecology, University of the Basque Country (UPV/EHU), Bilbao, Spain

* marius.dhamelincourt@inrae.fr

**Data Availability Statement:** Data and code to run the analysis are available in the INRAE dataverse repository (https://doi.org/10.15454/BEHOQK) under the name "Sea lamprey nests promote the

## Abstract

The habitat heterogeneity hypothesis states that increased habitat heterogeneity promotes species diversity through increased availability of ecological niches. We aimed at describing the local-scale (i.e. nest and adjacent substrate) effects of nests of the sea lamprey (*Petromyzon marinus* L.) as ecosystem engineer on macroinvertebrate assemblages. We hypothesized that increased streambed physical heterogeneity caused by sea lamprey spawning would modify invertebrate assemblages and specific biologic traits and promote reach-scale diversity. We sampled thirty lamprey nests of the Nive River, a river of the south western France with a length of 79.3 km and tributary of the Adour River, in three zones: the unmodified riverbed (upstream) and zones corresponding to the nest: the area excavated (pit) and the downstream accumulation of pebbles and cobbles (mound). The increased habitat heterogeneity created by lamprey was accompanied by biological heterogeneity with a reduced density of invertebrates (3777 ± 1332 individuals per $m^2$ in upstream, 2649 ± 1386 individuals per $m^2$ in pit and 3833 ± 1052 individuals per $m^2$ in mound) and number of taxa (23.5 ± 3.9 taxa for upstream, 18.6 ± 3.9 taxa in pit and 21.2 ± 4.5 taxa for mound) in the pit compared to other zones. However the overall taxa diversity in nest increased with 82 ± 14 taxa compared to the 69 ± 8 taxa estimated in upstream zone. Diversity indices were consistent with the previous results indicating a loss of $\alpha$ diversity in pit but a higher $\beta$ diversity between a pit and a mound than between two upstream zones, especially considering Morisita index accounting for taxa abundance. Trait analysis showed high functional diversity within zones with a reduced proportion of collectors, scrapers, shredders, litter/mud preference and small invertebrates in mound, while the proportion of "slabs, blocks, stones and pebbles" preference and largest invertebrates increased. Pit presented the opposite trend, while upstream had globally intermediate trait proportions. Our results highlight important effects on species and functional diversity due to habitat heterogeneity created by a nest-building species, what can ultimately influence food webs and nutrient processes in river ecosystems.

diversity of benthic macroinvertebrate assemblages".

**Funding:** Functioning was financed by Pôle Gestion des Migrateurs Amphihalins dans leur Environnement (https://www6.rennes.inrae.fr/u3e/ PRESENTATION/Organisation/Pole-MIAME). M.D. PhDs was financed by University of Pau and Pays de l'Adour (https://www.univ-pau.fr/fr/index.html) and UPV/EHU (https://www.ehu.eus/es/home). Field work used resources from the IE ECP Experimental Facility of the UMR Ecobiop (https:// www6.bordeaux-aquitaine.inrae.fr/ie-ecp-ecobiop). The funders had no role in study design, data collection and analysis, decision to publish, or preparation of the manuscript. Grant no. IT1471-22 from the Basque Government funded this study.

**Competing interests:** The authors have declared that no competing interests exist.

## Introduction

Habitat heterogeneity has been widely recognized as an important factor structuring biological assemblages [1, 2]. The habitat heterogeneity hypothesis [3] states that increased habitat heterogeneity promotes species diversity thanks to a greater number of available ecological niches. Li and Reynolds [4] defined heterogeneity in an operational way as "the complexity and/or variability of a system property in space and/or time". Heterogeneity can be considered at different scales, and depending on the scale considered, it has contrasted effects on ecological processes [5]. In the case of stream invertebrates, substrate heterogeneity can affect food distribution [6] and thus condition their local-scale (i.e. nest-scale) feeding and movements [7], whereas movements at larger scales can respond to drift if there are no available interstices or substrate for attachment.

In aquatic ecosystems, heterogeneity analyzes are often used to explain macroinvertebrate assemblages. Beisel et al. [8] found that the number of macroinvertebrate taxa in a given area (considering 0.5 to 4 m radius zones of substrate in a fourth-order stream) was higher in a heterogeneous environment with more substrate types and an elevated patchiness. Furthermore, the same study found that homogeneous environments are more prone to the dominance of few taxa, probably due to the lack of competition with taxa from neighbouring patches. At a very fine scale, Boyero [9] found a positive relationship between substrate heterogeneity and taxonomic diversity across 225 cm$^2$ samples collected within the same square meter of riffle. Substrate complexity appears to be a key driver of macroinvertebrate assemblage structure followed by current velocity and depth [10]. However, increased heterogeneity does not necessarily enhance macroinvertebrate diversity, at least in stream restoration projects [11].

Ecosystem engineering [12, 13] is defined as the change of availability of resources to other species due to physical changes in biotic or abiotic materials, directly or indirectly mediated by organisms. During the process they modify the habitats or create new ones. Effects of engineer species [14], alongside with the interaction between hydraulics and the substrate, are abiotic and biotic processes influencing heterogeneity. Together, they affect habitat heterogeneity, which in turn can affect ecosystem functioning [6]. Substrate disturbance by foraging fish is another cause of substrate heterogeneity, which may destabilize the substrate and cause an increased bedload transport during subsequent flood events [15].

Fish nesting, especially in species such as salmonids, which move large volumes of sediment and alter local bed morphology, is a biotic process that can increase riverbed substrate heterogeneity [16]. For migratory species with high nest density in the spawning grounds, the ecological effects can be important and cascade to affect biological assemblages and ecosystem processes. For instance, Moore and Schindler [17, 18] found that bed disturbance by salmon in spawning grounds caused severe seasonal declines in periphyton and benthic invertebrate abundance, with no recovery within the same season. However, other species such as chubs of the genus *Nocomis* build nests that promote macroinvertebrate density and even allow some taxa to persist in degraded streams [19]. Migratory species likely to affect substrate heterogeneity and the assemblages include the sea lamprey (*Petromyzon marinus* L.), which builds 40–220 cm-wide nests [20–22] in late spring ([23], Fig 1). Both male and female lamprey remove cobbles with their mouth and release them downstream so that the resulting nest consists of a pit carpeted with sand or gravel, followed by a mound of pebbles and cobbles. This structural heterogeneity might have functional consequences, as the density of some taxa such as Hydropsychidae, Philopotamidae and Heptageniidae can increase up to tenfolds in mounds [24] in an oligotrophic stream. Another study [25] found an increased Simuliidae density in areas with disturbed substrate and added sea lamprey carcasses, possibly resulting from the association of the cleared substrate with suitable characteristics and food supply from dead lampreys.

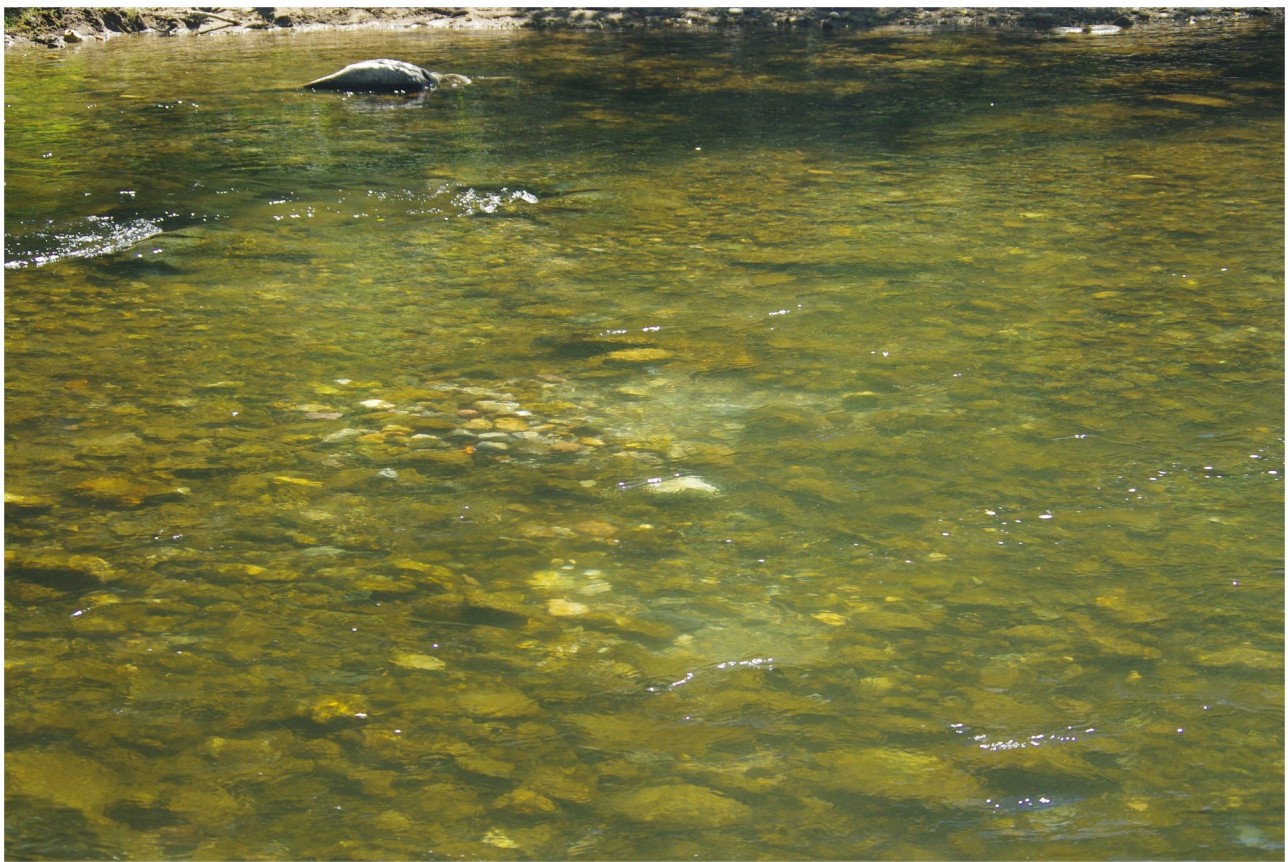

**Fig 1. Photo of two sea lamprey nests in the Nive River.** The mound of cobbles, immediately downstream of the pit, is clearly visible. ©INRAE-GLISE.

These results suggest an effect of sea lamprey nesting activity on macroinvertebrate assemblages, with potential functional consequences.

Here, we aimed at describing the local-scale (i.e. nest and adjacent substrate) effects of sea lamprey as ecosystem engineer on macroinvertebrate assemblages at the nest. We expect the taxonomic and functional diversity on the nest as a whole (pit+mound) to be higher than on undisturbed substrate. We hypothesized that increased streambed physical heterogeneity caused by sea lamprey nesting would promote invertebrate diversity and specific biologic traits. Pits, with deeper and more lentic conditions than unmodified riverbed (upstream) would attract lentophilic and fine sediment fauna, whereas the shallower and more lotic mounds would attract taxa with opposite preferences. In addition, nests with more heterogeneous current and depth may host more diverse communities. To test our hypotheses, we first assessed the variability of depth and current velocity on pit, mound and upstream as these variables are among the most likely to explain differences in macroinvertebrate assemblages [10]. Then, we compared the density, taxa richness and diversity of macroinvertebrates across nest zones. Finally, we studied ecological traits describing food type, mode of alimentation, substrate preference and body size, to infer the effects of nests on invertebrate functional heterogeneity.

## Methods

No permit was required to sample invertebrates in lamprey nests, because the sampling was performed after the larvae left the nests, thereby preventing disturbance to lamprey lifecycle.

### Study site and experimental period

The study took place in the Nive River, a 79 km long river situated in Northern Basque Country, France, and draining a basin of 1030 km$^2$ (river width: 18 m; bankfull width: 25 m; mean river depth: 52 cm; average discharge: 21 m$^3$/s). The selected section corresponds to a reach of 400 m long, located in Saint-Martin-d'Arrossa (43° 14' 34.926" N, 1° 18' 27.305" W) and bypassed by a hydropower scheme. It is mainly composed of riffles and runs suitable for sea lamprey nesting, as shown by the 46 nests found on a previous survey [26]. To avoid disturbing sea lamprey spawning, macroinvertebrate samples were collected on July 7 2020, at the end of the reproductive season, after two weeks of not detecting any lamprey near the nests. We considered these two weeks a period sufficient to allow for stability of local invertebrate assemblages [27], whereas reducing the risk of scouring by floods, which are frequent at the study site during late spring and early summer.

### Macroinvertebrate samples and nest characteristics

We studied 30 completed nests where no lamprey was observed (Fig 2). For each nest, we defined three zones. The **upstream zone**, 20 to 50 cm upstream from the upstream verge of the nest pit, indicated the initial conditions of the habitat chosen by the lamprey to build their nest, being both close to the nest and less likely to be disturbed by digging than the left, right

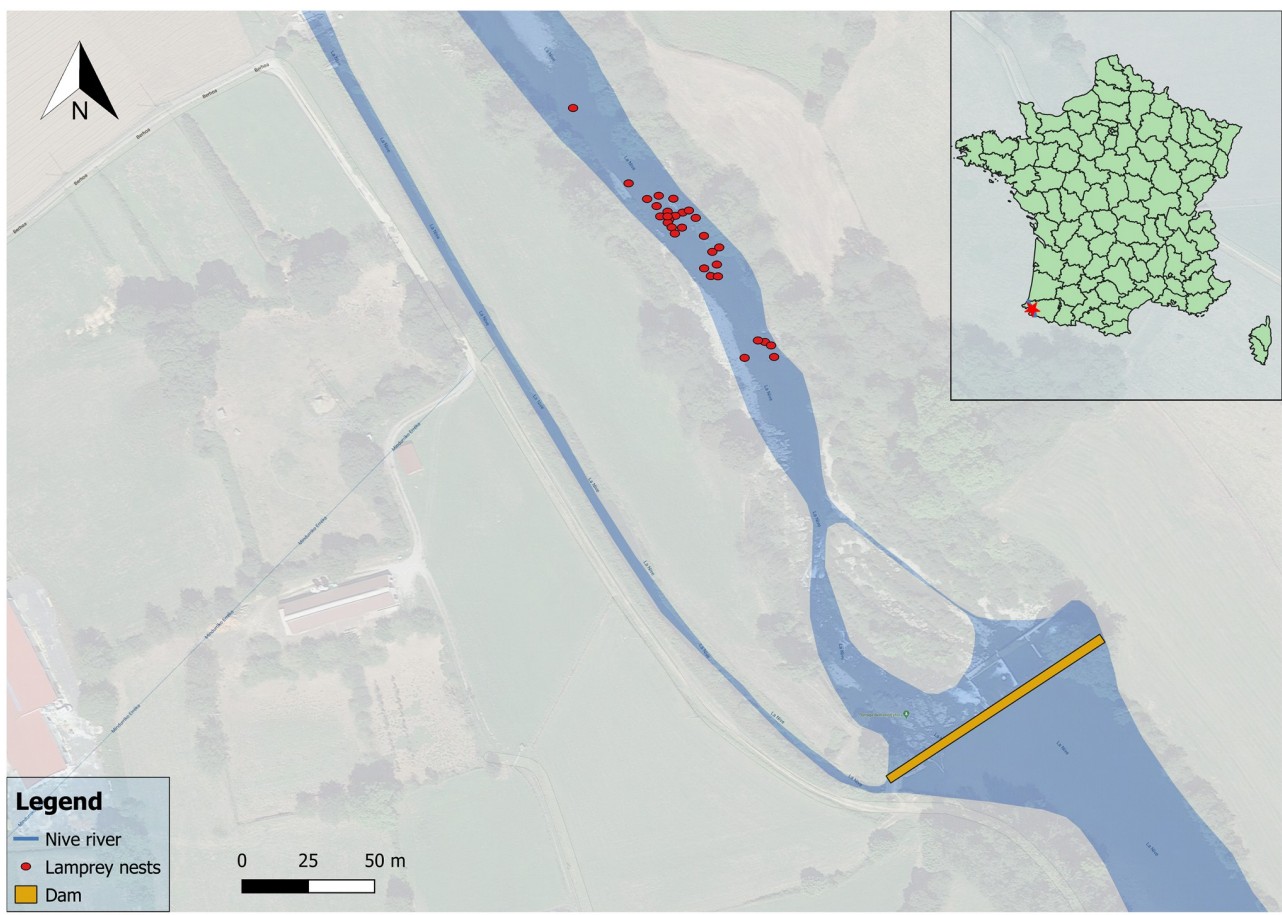

**Fig 2. Location of the nests sampled in the selected section of the Niver river in Saint-Martin-d'Arrossa, bypassed by a hydropower scheme.**

or immediate downstream of the nest. The second zone was the **pit**, delimited as the area excavated by the lamprey, where water was deeper and the substrate finer than upstream. The third zone was the **mound**, corresponding to the downstream accumulation of pebbles and cobbles. For all three zones current velocity (± 1 cm/s) was measured at 5 cm from the bottom (using a magnetic flowmeter "FLO-MATE 2000") and water depth (± 1 cm) was measured with a Vernier gauge. Transversal (perpendicular to the current) and longitudinal (parallel to the current) diameters of the pit were measured using a measuring tape (± 1 cm).

Macroinvertebrates were sampled using a Surber net (1/20 $m^2$, 0,5 mm mesh) by stirring the entire top layer of the sediment covered by the net frame (no more than 50 mm). Three samples, one per zone, were collected on each nest. After collection the samples were stored in 70˚ ethanol up to the determination. Most invertebrates were identified following AFNOR standard XP T 90–388 at the genus level except for Oligochaeta (subclass), Diptera (family) and some individuals for whom genus determination was not certain (determined at the family level). Macroinvertebrate traits were determined following Tachet et al. [28] and expressed as the frequency of individuals belonging to a certain trait over the total number of individuals in the sample. We selected food type, mode of alimentation, substrate preference and body size as these traits are likely to vary according to the physical characteristics of the nests. Food type and mode of alimentation directly reflect the resources provided by a given environment. With different current and substrate characteristics those traits should vary between zones. As substrate will differ, substrate preferences may reflect the sorting made by spawners between zones, if the scale of the nest has an impact on macroinvertebrate spatial location. Finally, body size emphasizes the carrying capacity for smaller or bigger invertebrates, which completes the substrate preference.

## Invertebrate diversity

All analyzes were performed using R software version 4.1.0 [29] and a significance level of 0.05.

We used several $\alpha$ and $\beta$ indices to measure the effects of lamprey nests on the diversity of macroinvertebrate assemblages. For $\alpha$ indices we calculated the indices in each zone to compare the pit and the mound with the upstream zone. First, we measured the species richness with the *specpool* function of the *vegan* package [30, version 2.5–7] using a Chao1 equation [31, Eq 1]. Then, Shannon $\alpha$ diversity index [32, Eq 2], calculated using the *diversity* function of the *vegan* package [30, version 2.5–7], indicated how diverse the taxa in each zone are, increasing with the increase of evenness and taxa richness in a sample. Shannon index was completed with Pielou index [33, Eq 3] to specifically analyze the evenness of the assemblages. These indices were transformed into Log Response Ratio (LRR), corresponding to the effect size of the diversity differences between nest (pit/mound) zones and upstream zones (Eq 4 and Fig 6) where the "index modified zone" corresponds either to the index of a pit or to the index of a mound. Indices were calculated for each pit and each mound and compared to the mean of Shannon and Pielou indices obtained for the upstream zones ("index upstream zone"). When the LRR standard deviation does not cross the value "0", it means that the effect is significantly positive (above 0) or negative (below 0; [34]). Behind this analysis, we compared how a given index (Shannon or Pielou) differed on a pit or on a mound—thus reflecting a disturbed habitat—compared to the average value of these indices in the section considered.

$$Chao\ 1 = S + \frac{a^2}{2b} \tag{1}$$

*Where S is the number of taxa in a sample, a the number of taxa solely represented by one individual in the sample, and b the number of taxa represented by exactly two individuals in the sample.*

$$Shannon\ index(H') = -\sum_{i=1}^{S} p_i \ln p_i \tag{2}$$

*Where $p_i$ is the proportion of a taxon i on the total number of individuals N and S is the total number of taxa in the sample.*

$$Pielou\ evenness\ index(J') = \frac{H'}{H'_{max}} \tag{3}$$

*Where H'is the Shannon diversity index and $H'_{max}$ is the maximum possible value of H'(if every species was equally likely): $H'_{max} = -\sum_{i=1}^{S} \frac{1}{S} \ln \frac{1}{S}$ and S being the total number of taxa in the sample.*

$$Log\ Response\ Ratio = \ln\left(\frac{index\ modified\ zone}{index\ upstream\ zone}\right) \tag{4}$$

Following this $\alpha$ diversity overview we studied several $\beta$ diversity aspects to compare nest and upstream. For each index we calculated all the pairwise differences between pit and mound and between two upstream zones. Firstly, we calculated Sorensen dissimilarity index [35, Eq 5] which includes both replacement and nestedness and corresponds to a global indicator of taxa diversity. Then we calculated Simpson $\beta$ diversity index to study taxa replacement [35, Eq 6]. The second component of the Sorensen index, the taxa nestedness, was obtained by computing the difference between Sorensen and Simpson indices [36]. Finally, to test the diversity considering taxa abundance and not solely their occurrence, we calculated the Morisita dissimilarity index [37, Eq 7]. As for the $\alpha$ diversity indices the $\beta$ indices were transformed into Log Response Ratio (Eq 4 and Fig 7) where the "index modified zone" corresponds to a pairwise index between a pit and a mound and the "index upstream zone" to a pairwise index between two upstream zones. This LRR indicates if the diversity difference between two locations of a nest (always a pit versus a mound) differs from the same comparison made between two upstream zones, thus reflecting a difference in taxa heterogeneity. To avoid a nest effect we did not compare the taxa of a pit versus the mound of the same nest and therefore we randomly selected the pit and the mound to be compared in each index value. Indeed, pit and mound of the same nest may not be independent, and this analysis aimed at describing the heterogeneity of communities between pits and mounds in general, without the influence of similarities existing between adjacent samples.

$$Sorensen\ \beta\ diversity\ index = \frac{(b+c)}{(2a+b+c)} \tag{5}$$

*Where a is the number of common taxa between two sites, b the number of taxa exclusive to a site, and c the number of taxa exclusive to the other site.*

$$Simpson\ \beta\ diversity\ index = \frac{\min(b,c)}{\min(b,c) + a} \tag{6}$$

*Where min (b, c) is the lowest number of exclusive taxa between two sites, b being the number of taxa exclusive to a site, and c the number of taxa exclusive to the other site; a is the number of*

*common taxa between the two sites.*

$$Morisita\ overlap\ index(C_D) = \frac{2\sum_{i=1}^{S} x_i y_i}{(D_x + D_y)XY} \tag{7}$$

*Where $x_i$ is the number of times taxon i is represented in the total X from one site; $y_i$ is the number of times taxon i is represented in the total Y from another site; $D_x$ and $D_y$ are the Simpson's α diversity index for the x and y sites respectively:*

$$D_i = \sum_{i=1}^{S} p_i^2 \tag{8}$$

*With S the total number of taxa in the sample and $p_i$ the proportion of a taxon i on the total number of individuals N.*

## Data analysis

The significance of differences between nest zones for depth, current, density, diversity, Shannon index and Pielou index was determined using linear mixed model and the *lmer* function of the *lme4* package [38] with a random effect of nest identity.

For density and taxa diversity (Fig 4A and 4B), a pairwise Tukey test was used to check the differences between each zone rather than globally.

To test the effect of current and depth on abundance and taxa diversity, a General Linear Model was used with current and depth as covariates and following a Gaussian distribution.

A two-sample t-test was used to test the significance of Chao1 diversity index calculated for nest (pit + mound) and upstream zone.

Trait differences between zones were determined using binomial mixed models with upstream or mound as reference condition and nest identity as a random effect (function *glmer* of the *lme4* package; Bates et al. [38].

The links between trait and nest characteristics were studied using Redundancy analysis (RDA), a multivariate approach of linear models applied in *vegan* package [30, version 2.5–7] with *rda* function and setting traits as response variables. Traits were implemented after Hellinger transformation to give low weight to traits with low counts and many zeros. Explanatory variables were the five variables describing nest characteristics: transversal and longitudinal diameter, depth, current and depth difference between pit and mound. The latter was calculated assuming nests with higher depth difference could be more heterogeneous with a more important volume and habitat complexity.

## Results

### Nests characteristics

Lamprey nests differed greatly in their dimensions, transversal diameter of the pit averaging 129.2 ± 37.2 cm (range, 70–210 cm) and longitudinal diameter 117.4 ± 45.7 cm (range, 50–260 cm, Table 1). The pit was the deepest zone (59.2 ± 9.9 cm) followed by the upstream (51.8 ± 10.1 cm) and the mound (38.3 ± 10.1 cm), and differences were statistically significant considering mixed models p-values (Table 2). The coefficient of variation for depth was similar for all three zones (Table 2). Current velocity ranged from 0 to 65 cm/s (Table 1) and varied among zones inversely to depth (Fig 3), although the coefficient of variation was larger. Again, differences were statistically significant considering mixed models results.

**Table 1. Summary of the nests characteristics measured in the study; mean ± sd (cv).**

|  | Pit | Mound | Upstream |
|---|---|---|---|
| Transversal diameter (cm) | 129.2 ± 37.2 (0.3) | - | - |
| Longitudinal diameter (cm) | 117.4 ± 45.7 (0.4) | - | - |
| Depth (cm) | 59.2 ± 9.9 (0.2) | 38.3 ± 10.1 (0.3) | 51.8 ± 10.1 (0.2) |
| Current (cm/s) | 8.8 ± 7.8 (0.9) | 43.8 ± 11.6 (0.3) | 19.8 ± 11.1 (0.6) |
| Depth difference (cm) | 20.9 ± 4.8 (0.2) | | - |

## Macroinvertebrate density and diversity

The density of macroinvertebrates (Fig 4A) was 2649 ± 1386 individuals per $m^2$ in pit, 3833 ± 1052 individuals per $m^2$ in mound and 3777 ± 1332 individuals per $m^2$ in upstream. It was significantly lower in pit than in upstream and mound (*Tukey test: Df = 58, t.ratio = 3.529, p-value = 0.0023* and *Df = 58, t.ratio = -3.705, p-value = 0.0014* respectively) but there were no differences between upstream and mound (*Tukey test: Df = 58, t.ratio = -0.175, p-value = 0.9832*). The taxa richness (Fig 4B) also did not vary significantly between upstream (23.5 ± 3.9 taxa) and mound (21.2 ± 4.5 taxa; *Tukey test: Df = 58, t.ratio = 2.293, p-value = 0.0647*) but was higher in these two zones than in pit (18.6 ± 3.9 taxa; *Tukey test: Df = 58, t.ratio = 5.066, p-value = ≤ 0.0001* for pit compared to upstream and *Tukey test: Df = 58, t.ratio = -2.772, p-value = 0.0201* for pit compared to mound).

General Linear Models indicated that current affected neither the number of taxa nor the macroinvertebrate density (*tvalue = 0.391; Pr(> |t|) = 0.697* and *tvalue = 1.470; Pr(> |t|) = 0.1452* respectively). Depth influenced negatively the macroinvertebrate density (*tvalue = −2.026; Pr(> |t|) = 0.0458*) but not the number of taxa (*tvalue = −0.821; Pr(> |t|) = 0.414*).

The Venn diagram (Fig 5), describing exclusivity or sharing of taxa, indicates an important taxa overlap with 59% of the taxa found on all zones. The significant pattern of reduced taxa richness found at the nest scale was not found by summing all the different taxa identified in the thirty samples of each zone and visualizing it in the Venn diagram.

## Diversity

The estimate of Chao1 diversity index was 82 ± 14 taxa for nest (pit + mound) and 69 ± 8 taxa for the upstream zone. In spite of the slight overlap of the standard error of those estimates the species richness was higher in the nest than upstream, as supported by results of two-sample t-test (*Df = 44.739, t = 4.5017, p-value ≤ 0.0001*). Log response ratios of the $\alpha$ diversity indices (Fig 6) showed an overall trend of reduced diversity and equitability for both Shannon and Pielou indices in pit and mound, compared to the average values observed in the upstream zone.

**Table 2. Mixed model results for analyzes of differences between zones for nest characteristics, density and diversity of macroinvertebrates and $\alpha$ diversity indices, with nest identity as a random effect.** With $^*P \leq 0.05$, $^{**}P \leq 0.01$, $^{***}P \leq 0.001$.

| Variables | Df | $\chi^2$ | P |
|---|---|---|---|
| Depth | 2 | 503.5 | ≤ 0.001*** |
| Current | 2 | 278.41 | ≤ 0.001*** |
| Density | 2 | 17.475 | ≤ 0.001*** |
| Diversity | 2 | 25.736 | ≤ 0.001*** |
| Shannon index (upstream/pit/mound) | 2 | 12.739 | 0.0017 ** |
| Pielou index (upstream/pit/mound) | 2 | 6.6937 | 0.035 * |

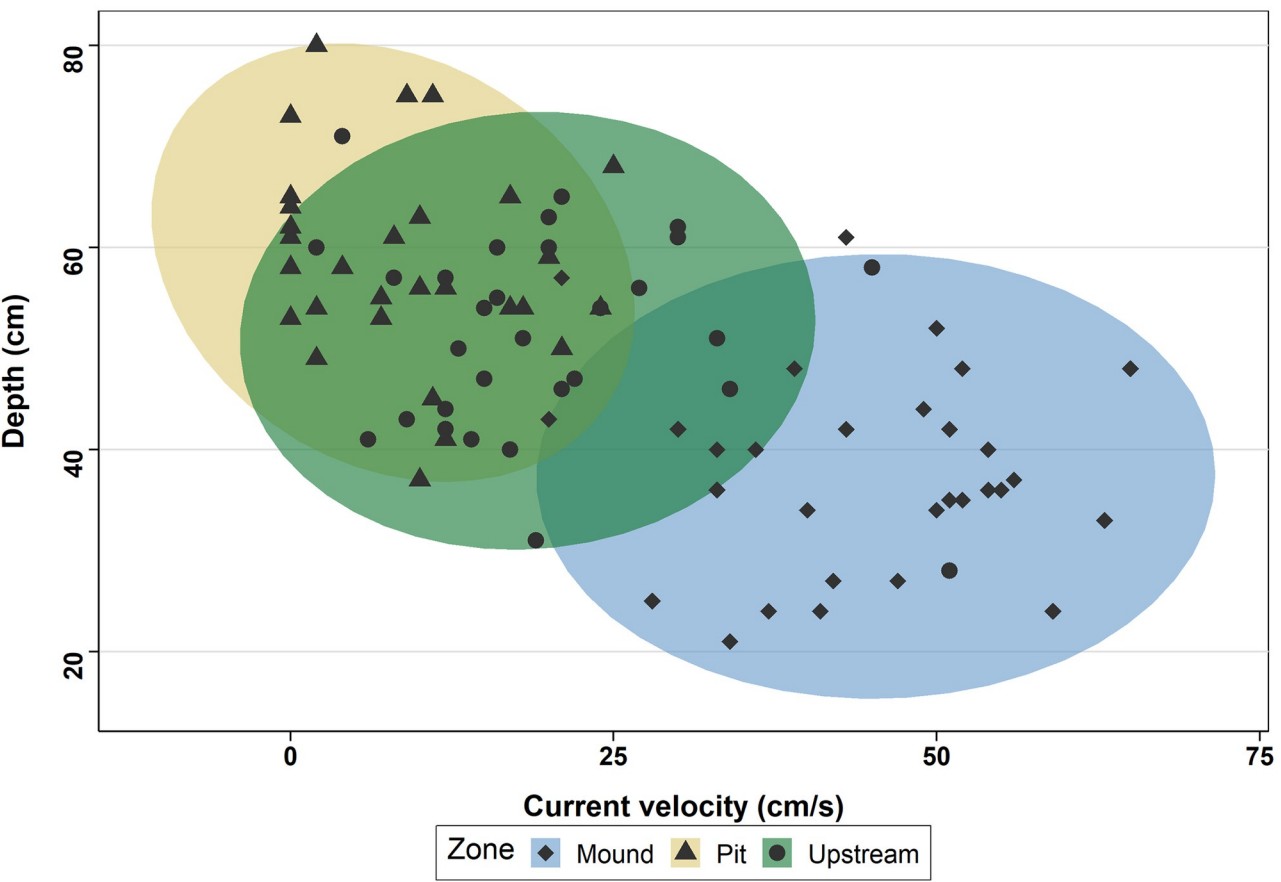

**Fig 3. Relationship between depth and current velocity for each nest at each zone.** Ellipses correspond to multivariate t-distribution.

However, only the log response ratio of the Shannon index of the pit is strictly below 0 and indicates a significantly reduced diversity in pit compared to the upstream zone (-10.5 ± 10.2%). The log response ratios of $\beta$ diversity indices (Fig 7) showed a higher overall $\beta$ diversity between a pit and a mound than between two upstream zones. However the log response ratios were highly variable and none were significantly different from 0. Nevertheless the higher Morisita index ratio compared to other indices seems to indicate a differentiation made by taxa abundance rather than by replacement or nestedness.

## Trait analysis

The traits studied tended to be more similar in mound and upstream than between these two locations and the pit (Table 3 and Figs 8–10). Considering alimentation traits (Fig 8), the proportion of collectors and scrapers was lower in the mound than in the pit and the proportion of filterers was lower in pit than either in mound or upstream. Predators were more present in mound than in pit and upstream, whereas shredders were less abundant in mound than in pit and upstream. However, food traits, related to alimentation aspects, did not highlighted significant differences. Substrate preference analysis (Fig 9) showed a similar pattern for litter and mud with higher proportion in pit than in mound. Preference for gravel and sand was more important in pit, followed by upstream and then mound. Finally, preference for slabs, blocks, stones and pebbles was highest in mound and lowest in pit. The smallest size category (Fig 10)

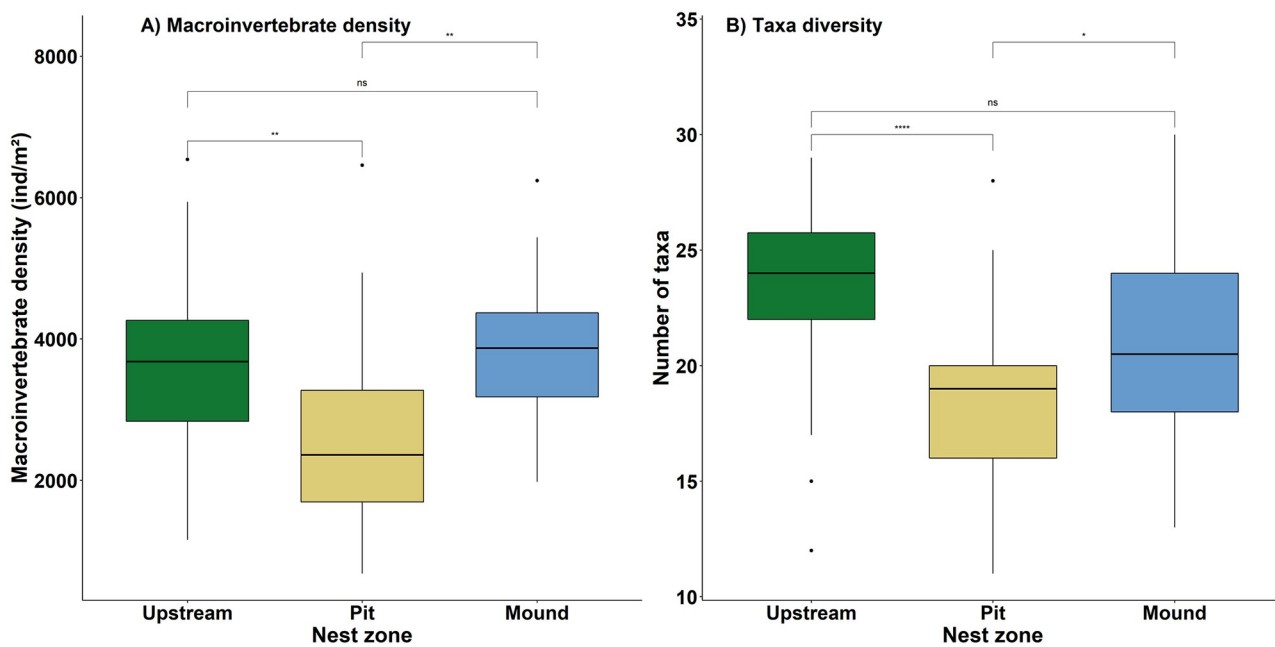

**Fig 4. Density of macroinvertebrates ($ind/m^2$) (A) and number of taxa per sample (B) for the three zones studied.** With *ns* $P > 0.05$, $^*P \leq 0.05$, $^{**}P \leq 0.01$, $^{****}P \leq 0.0001$.

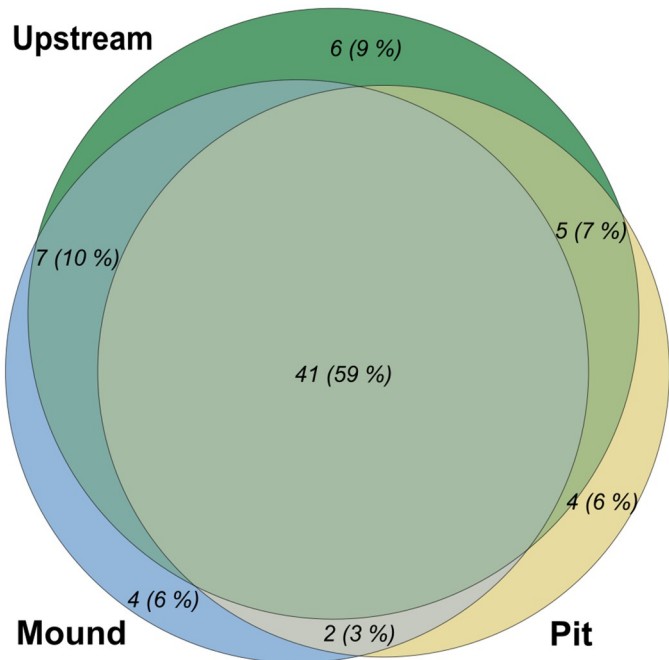

**Fig 5. Venn diagram with the distribution of taxa among zones.** The area of each part is proportional to the number of taxa indicated in absolute number and percentage of total taxa richness.

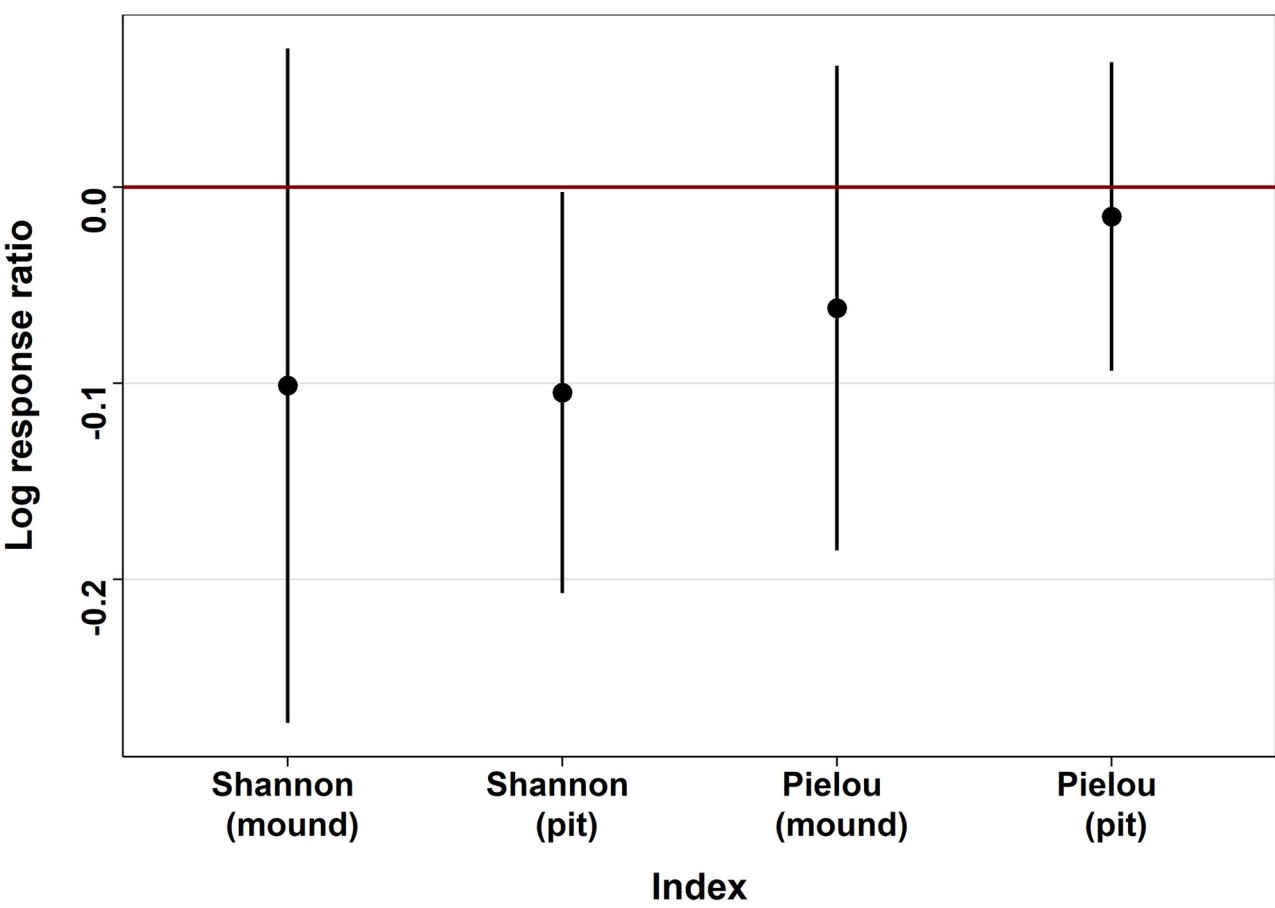

**Fig 6. Log response ratios of Shannon and Pielou $\alpha$ diversity indices for each pit and each mound compared to the mean of Shannon and Pielou indices obtained for the upstream zones.** Dots correspond to the mean and error bars correspond to the standard deviation. When the LRR standard deviation does not cross the value "0", it means that the effect is significantly positive (above 0) or negative (below 0).

was more represented inside the pit, followed by upstream and mound, whereas larger size categories followed the inverse pattern.

### Relationship between traits and nest characteristics

Redundancy analysis (Fig 11) yielded a significant model ($F = 10.876$, $Pr(> F) = 0.001$), although the only significant term was the current ($F = 46.8357$, $Pr(> F) = 0.001$), depth and depth difference being marginally insignificant ($F = 2.4535$, $Pr(> F) = 0.077$ and $F = 2.6966$, $Pr(> F) = 0.077$ respectively). The ellipses indicated a separation of pit from mound and upstream structured by current and "< 5 mm", "scrapers", "sand" and "gravel" trait categories belonging to the pit ellipse. Upstream and mound were not clearly separated but "5–10 mm", "slabs, blocks, stones and pebbles" and "filterers" belong to the mound ellipse. Those results are consistent with the specific trait analyzes (Figs 8–10). Depth, if not the most discriminant variable, explained the separation of pit from other zones.

### Discussion

Firstly, our results globally showed that habitat diversity and structural heterogeneity due to lamprey nesting activities modify several aspects of biodiversity. The clear distinction between

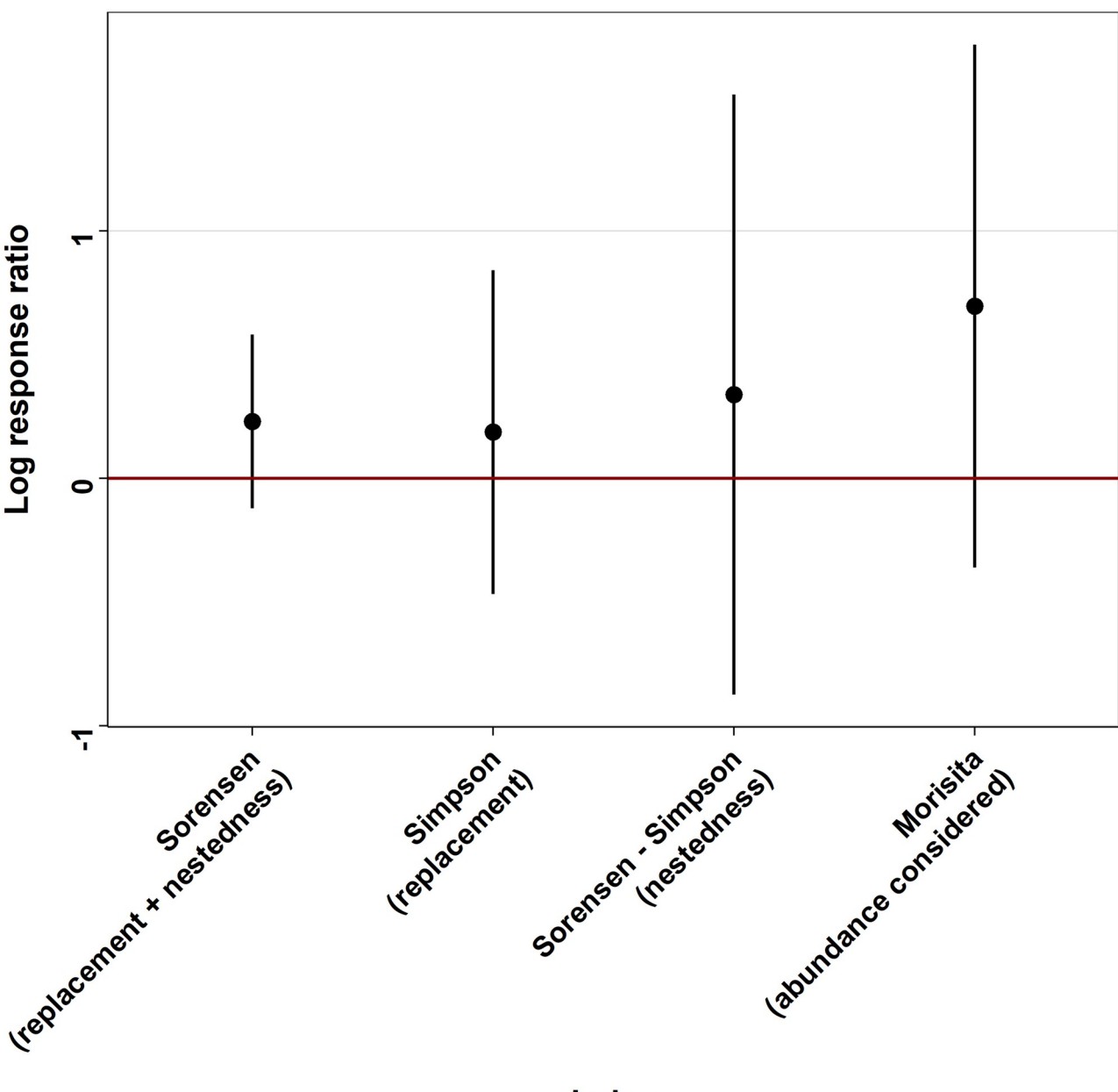

**Fig 7. Log response ratios of β diversity indices of all mound and pit pairwise indices, each compared to a pairwise upstream index among all possible upstream pairwise comparisons (upstream zone randomly selected, not directly upstream from the pit and mound considered).** Dots correspond to the mean and error bars correspond to the standard deviation. When the LRR standard deviation does not cross the value "0", it means that the effect is significantly positive (above 0) or negative (below 0).

nest zones regarding depth and riverbed current velocity implied biological heterogeneity, a non trivial result as previously highlighted [11]. The pit had a lower density of invertebrates and a lower number of taxa than the other zones, which did not differ between them, although the number of taxa tended to be higher upstream. Log response ratio of $\alpha$ Shannon index showed a similar trend concerning the diversity inside the pit. Those results indicate that nest building reduced the local abundance and diversity of macroinvertebrates in the pit, at least

**Table 3. Binomial mixed models results comparing frequency of selected traits between zones with nest identity as a random effect.** With *$P \le 0.05$, ***$P \le 0.001$.

|  | Zone | Df | Z value | P |
|---|---|---|---|---|
| Food | Mound (reference: upstream) | 2 | -1.016 | 0.31 |
|  | Pit (reference: upstream) | 2 | -0.614 | 0.539 |
|  | Mound (reference: pit) | 2 | -0.401 | 0.688 |
| Alimentation | Mound (reference: upstream) | 2 | 1.877 | 6.05e-2 |
|  | Pit (reference: upstream) | 2 | -2.201 | 2.77e-2 * |
|  | Mound (reference: pit) | 2 | 4.063 | 4.84e-5 *** |
| Substrate | Mound (reference: upstream) | 2 | -6.918 | 4.57e-12 *** |
|  | Pit (reference: upstream) | 2 | 10.779 | <2e-16 *** |
|  | Mound (reference: pit) | 2 | -17.391 | <2e-16 *** |
| Size | Mound (reference: upstream) | 2 | 5.682 | 1.33e-08 *** |
|  | Pit (reference: upstream) | 2 | -11.63 | <2e-16 *** |
|  | Mound (reference: pit) | 2 | 16.71 | <2e-16 *** |

during the first weeks after nest digging, which are the most significant from the point of view of the lamprey, as larvae abandon the nest shortly after hatching [39]. Patches of fine substrate such as the pit tend to be less diverse than zones with coarser grain size [40]. Reduced invertebrate abundance and richness have been also reported in nests of the largemouth bass (*Micropterus salmoides*) [41] and the pink salmon (*Oncorhynchus gorbuscha*) [42] during the spawning season, which was attributed to spawning-related disturbance. On the other hand, Hogg et al. [24] reported that the density of invertebrates in sea lamprey nest mounds were twice that found in pits and 77% higher than in upstream zone. Their results of reduced macroinvertebrate pit abundance are consistent with ours but we did not observe an increased mound density. An explanation for the important density in mounds could be the dominance of the Chironomidae within the river studied, not found for the Nive River (S1 Fig). The difference between our results and theirs seems not to derive from differences in macroinvertebrate taxa composition, as it was relatively similar between both studies (with Chironomidae, Hydropsychidae, Heptageniidae and Ephemerellidae dominant in [24] and abundant in the Nive River: S1 Fig and S1 Table. The significantly higher number of taxa in nest (mound and pit combined) than upstream, highlighted by the Chao1 index, indicates that sea lamprey nests create heterogeneity but also increase local species diversity.

In a local scale, i.e. the zone, we found for each pit and mound several taxa not present upstream (see S2 Fig). Indeed, whereas upstream is more complex and offers more substrate types with its unsorted grain size (due to the absence of sorting by lamprey during the nest building compared to the nest)—likely to provide suitable habitat for more taxa— [43], mound and pit provide larger areas of some specific characteristics, including substrate. This wider area may potentially allow the establishment of more taxa sharing similar ecological preferences, when these would be less represented and therefore potentially absent or very rare upstream. Considering the nest scale, rare taxa should be found more easily, either in mound or in pit considering their preferences than upstream (considering a same sample size).

In addition to increasing the structural and biological heterogeneity, sea lamprey nests seem to shape invertebrate assemblages. Indeed, three traits over the four studied showed differences between nest and the upstream zones, although the direction of change did not always follow our expectations. In particular, we expected herbivores to increase in mounds, as epilithic algae tend to be favored by shallow water, coarse sediments and moderately fast flow [44, 45], but no significant difference was found globally for alimentation traits, perhaps because of

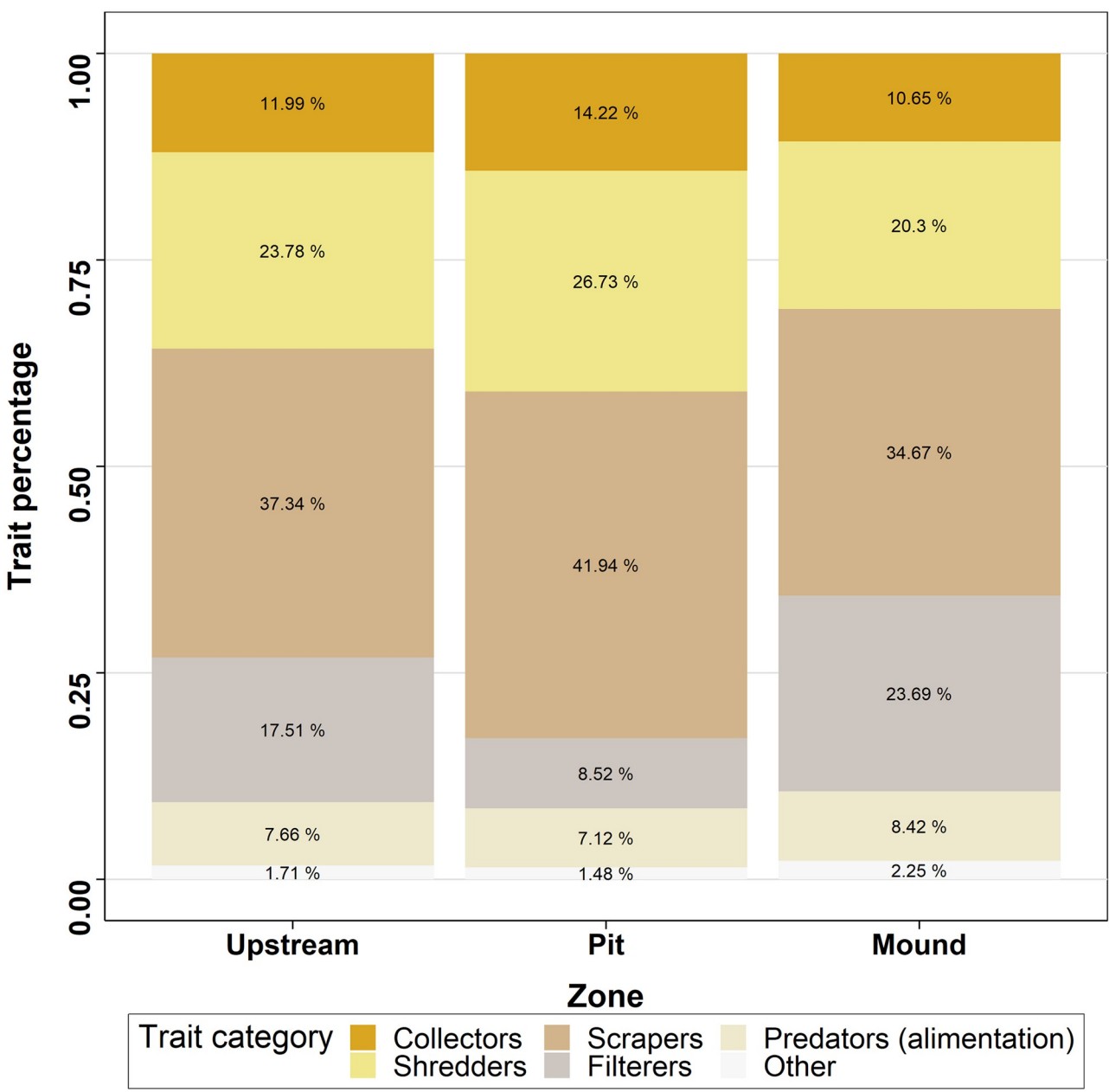

**Fig 8. Median percentage of macroinvertebrate alimentation traits for the three zones studied.**

lack of time for algae to grow on overturned cobbles. The increased proportion of collectors in pits was expected, as these are a preferential place for organic matter deposits. Similarly, collectors tend to be more abundant in fine substrate, which collected more detritus [46]. The higher concentration of organic matter might also explain the higher abundance of shredders and scrapers in pit than in mound. Also, according to our expectations, filterers were more represented in the mound, as they are favoured by fast-flowing areas [47], particularly during low flow conditions [48]. Those results reflect the different dynamics of food provided within the nest zones.

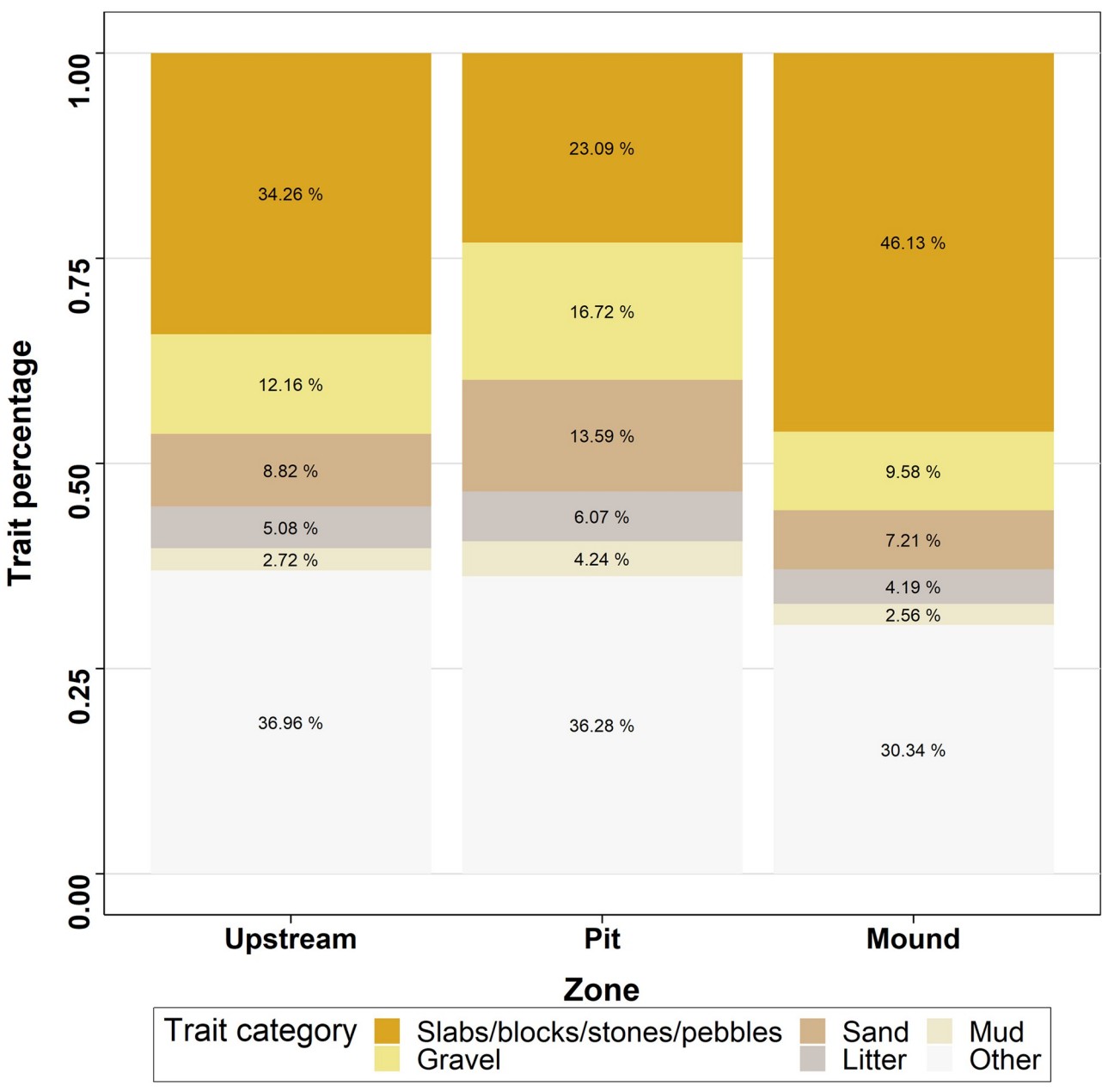

**Fig 9. Median percentage of macroinvertebrate substrate preferences for the three zones studied.**

Not surprisingly given the differences in substrate among zones, invertebrates also differed in their preferred substrate traits. Macroinvertebrates associated with coarse substrate were especially abundant in mounds, whereas taxa associated with fine substrate were more abundant in pits. It must be noted that we visually noticed an absence of sand in the substrate surface of the study reach, apart from lamprey nests and some marginal areas, which suggests that nesting lampreys create patches of habitat for lentophylic species in river stretches where these species could not dwell otherwise.

The macroinvertebrate size traits showed a greater proportion of small invertebrates (< 5 mm) within the pit, including *Esolus sp.*, the most abundant taxon in pit (S1 Fig) and

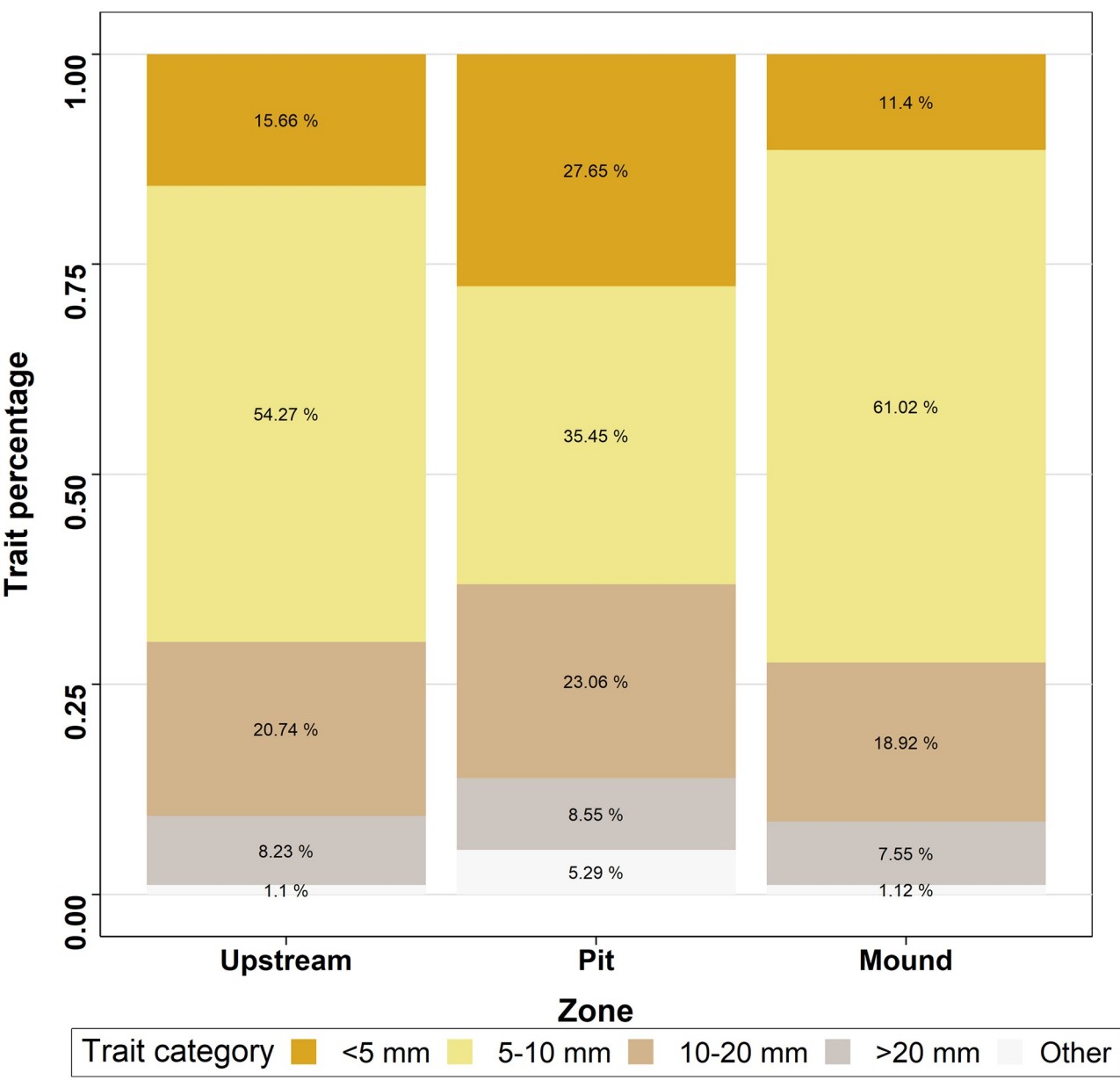

**Fig 10. Median percentage of macroinvertebrate size traits for the three zones studied.**

*Psychomyia pusilla*. This result could be explained by the carrying capacity of fine substrate. A fine substrate has less shelters than a coarser one. Shelters of fine substrate are more suitable for small size range of macroinvertebrates, whereas bigger macroinvertebrates are more prone to find shelter in the coarser substrate of the mound or upstream. Bêche et al. [49] suggested a better exploitation of refugia for small sized macroinvertebrates. Data on the grain size distribution of nest zones may help to confirm this hypothesis. Furthermore, small body size could be a possible resistance trait to disturbances such as riverbed movement [50].

Our results demonstrate that sea lamprey creates physical heterogeneity, which then enhances biological heterogeneity both in assemblage composition and function. The

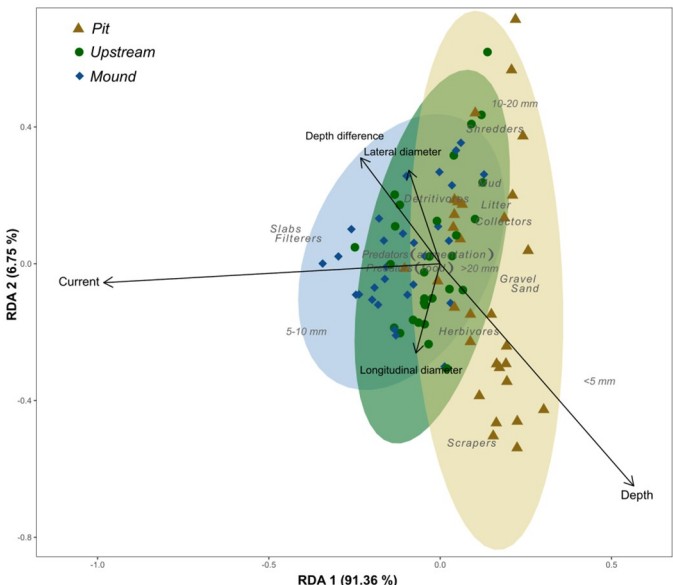

**Fig 11. Redundancy analysis between nest characteristics and macroinvertebrate traits.** Slabs = *slabs, blocks, stones and pebbles.*

heterogeneity demonstrated in this study was created at the nest scale, but local-scale processes can have a major impact at higher scales such as the reach scale [51]. In the case of lamprey nests in the Nive River, pits average 1.15 $m^2$ and so the total modified streambed area averages 34.5 $m^2$, representing 4% of the river streambed in our studied zone. With an average reduction of 30% of macroinvertebrate density in pit (compared to upstream) a total reduction of 1.2% of density is expected in the studied zone. In a river reacting in a similar way to the Nive, this surface with decreased macroinvertebrate density does not seem to be compensated by a higher density in the mound and so, sea lamprey nests globally decrease the macroinvertebrate density. To complete this finding, samples collected on a substrate further from the nests than our upstream samples may help to verify than they were not negatively affected by the downstream digging activity of lampreys.

Measuring the effect of lamprey nests on reach-scale macroinvertebrate density would require a BACI design [52] including periods and reaches with and without nests. If invertebrates affected by lamprey nesting responded with small-scale movements, the result would be an increase in macroinvertebrate density in upstream samples of reaches with nests compared to upstream samples of reaches without nests. If the macroinvertebrates responded drifting, the results would be more complex to interpret and depend on the distribution of nests of sea lamprey. An increase of macroinvertebrate densities could be observed if spawning sites occur upstream in the river. Macroinvertebrates drifting from those nests could colonize and increase the density of "upstream" sites situated downstream. When a spawning ground is located on the upper limit of the area colonized by lampreys, drift will not affect the adjacent area of this spawning ground but may affect downstream sites.

Our results highlight the need for a better understanding of the effects of nest-building species on the river ecosystem. Unlike salmon, whose effects are relatively well-studied [53–55], the consequences of sea lamprey spawning are still poorly known. Whereas lamprey nests only represent 4% of the river streambed in our study, the occupied area may be much higher, especially for invasive populations of the Great Lakes [56] or below impassable dams, likely to

dramatically increase the number of spawners in immediately downstream spawning grounds [26, 57]. Studies exist on the effects of carcasses [25, 58] or substrate modification [24] on a local scale, but not at the river or at the global scales. However, macroinvertebrates have a basal to intermediate position on the food chain [59]. Such position implies an important relationship with ecosystem components. Firstly, macroinvertebrate assemblages can have an important influence on numerous processes [59], such as nutrient cycles [60–62], primary productivity [63], decomposition [64–66], and translocation of materials [67, 68]. Then, they are an important resource for organisms belonging to higher trophic levels such as fish species. Macroinvertebrates are considered as the most important source of food, widespread in all freshwater ecosystems [69].

In our study some trait aspects can provide clues about the effects of sea lamprey on general ecosystem functioning in the river. Increased proportion of shredders in the pit suggests they find suitable conditions such as increased plant debris. On a more global scale a certain amount of material (depending on the nest density and count) could be retained by nests in spite of being carried away by the current, and then consumed by macroinvertebrates. Seeing this impact in late spring when litter input is reduced also indicates that the proportion of shredders should be more important in autumn, where these inputs can represent 73% of annual allochtonous inputs (in temperate deciduous forests; Abelho and Graça [70]). But this retention can occur only if the lamprey nests remain in autumn, without being filled by floods. Changes in streambed complexity created by nests persist during the autumn [24] but a change in hydrological conditions is likely to affect this process. Filterers are another trait category possibly affecting organic matter processing, being one pathway through which carbon and nutrients are transferred from the water column to the sediments [71]. Filterers being more present in proportion in the mound than upstream suggests that mounds are hotspots of nutrient processing. Therefore, sea lamprey nests could have an important effect on nutrient spiraling in streams, but the persistence of their influence on ecosystem functioning until the next spawning season remains to be assessed. Although nests may persist considering their physical structure, it is likely that the subsidies provided by eggs affect macroinvertebrate colonization dynamics and the distribution of functional groups, as predation was depicted in other species [72, 73]. An assessment of the effects of eggs deposited in the pit on functionality should be done to determine what is their potential contribution to the observed differences.

All these possible different effects imply that the modifications of macroinvertebrate assemblages by sea lampreys highlighted here and in previous studies [24, 25] are likely to modify the general functioning of the rivers. However, it is difficult to determine these consequences precisely due to the complexity inside macroinvertebrate assemblages. Such assessment requires specific studies on the components previously described. We advocate for studies focusing on these effects, important in a context of endangered native populations, likely to disappear in some rivers and consequently modifying their dynamics.

## Supporting information

**S1 Fig. Median of the abundances for the 10 most abundant taxa within each zone (Upstream, Pit, Mound).**
(TIFF)

**S2 Fig. Number of exclusive taxa in mound or pit compared to the upstream.** With $^{**}P \leq 0.01$.
(TIFF)

**S1 Table. List of taxa found in each zone of each nest.** With, for column names: *1 Leuctra; 2 Perlodes; 3 Perla; 4 Rhyacophila; 5 Glossosomatidae; 6 Agapetus; 7 Hydroptila; 8 Hydropsychidae; 9 Hydropsyche; 10 Polycentropodidae; 11 Polycentropus; 12 Psychomyiidae; 13 Psychomyia; 14 Oligoplectrum; 15 Micrasema; 16 Goeridae; 17 Lepidostoma; 18 Athripsodes; 19 Ceraclea; 20 Sericostomatidae; 21 Baetis; 22 Oligoneuriella; 23 Heptageniidae; 24 Epeorus; 25 Rhithrogena; 26 Ecdyonurus; 27 Ephemerella ignita; 28 Caenis; 29 Ephemera; 30 Potamanthus; 31 Hydraena; 32 Stenelmis; 33 Elmis; 34 Esolus; 35 Oulimnius; 36 Limnius; 37 Normandia; 38 Micronecta; 39 Aphelocheirus; 40 Blephariceridae; 41 Limoniidae; 42 Simuliidae; 43 Tanypodinae; 44 Ceratopogonidae; 45 Empididae; 46 Athericidae; 47 Gammaridae; 48 Echinogammarus; 49 Hydracarina; 50 Piscicolidae; 51 Erpobdellidae; 52 Oligochaeta; 53 Theodoxus; 54 Potamopyrgus; 55 Radix; 56 Planorbidae; 57 Ancylus; 58 Dugesiidae; 59 Nematoda; 60 Hydroporinae; 61 Hydrophilinae; 62 Prostoma; 63 Hydrozoa; 64 Ostracoda; 65 Copepoda; 66 Agapetinae; 67 Physella; 68 Leuctra geniculata; 69 Chironomidae excl. Tanypodinae.*
(PDF)

## Acknowledgments

Field work used resources from the IE ECP Experimental Facility of the UMR Ecobiop [74].

## Author Contributions

**Conceptualization:** Marius Dhamelincourt, Cédric Tentelier, Arturo Elosegi.

**Data curation:** Marius Dhamelincourt, Jacques Rives, Marie Pons, Cédric Tentelier, Arturo Elosegi.

**Formal analysis:** Marius Dhamelincourt, Marie Pons, Aitor Larrañaga.

**Funding acquisition:** Cédric Tentelier.

**Investigation:** Marius Dhamelincourt, Jacques Rives, Cédric Tentelier, Arturo Elosegi.

**Methodology:** Marius Dhamelincourt, Jacques Rives, Cédric Tentelier, Arturo Elosegi.

**Supervision:** Cédric Tentelier, Arturo Elosegi.

**Validation:** Cédric Tentelier, Arturo Elosegi.

**Writing – original draft:** Marius Dhamelincourt.

**Writing – review & editing:** Marius Dhamelincourt, Aitor Larrañaga, Cédric Tentelier, Arturo Elosegi.

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
