## [Decision Letter · Decision Letter 0]

24 Oct 2022

PONE-D-22-24495Sea lamprey nests promote the diversity of benthic macroinvertebrate assemblagesPLOS ONE

Dear Dr. Dhamelincourt,

Thank you for submitting your manuscript to PLOS ONE. After careful consideration, we feel that it has merit but does not fully meet PLOS ONE’s publication criteria as it currently stands. Therefore, we invite you to submit a revised version of the manuscript that addresses the points raised during the review process. This is an interesting paper exploring how habitat heterogeneity may promote species diversity. The focus on sea lamprey breeding sites to explore this idea is excellent. Generally, the data presented are robust. However, both reviewers point out areas which need attention to improve the overall impact of this work. I agree with their comments, especially related to explaining and justifying the experimental design (i.e., controls). Additionally, there are section which need additional details to ensure the paper is coherent across all sections.

We look forward to receiving your revised manuscript.

Kind regards,

Michael A Chadwick, PhD

Academic Editor

PLOS ONE

Journal Requirements:

"Functioning was financed by Pôle Gestion des Migrateurs Amphihalins dans leur Environnement. M.D. PhDs was financed by Univ. Pau and Pays Adour and UPV/EHU. Field work used resources from the IE ECP Experimental Facility of the UMR Ecobiop [71]."

"Functioning was financed by Pôle Gestion des Migrateurs Amphihalins dans leur Environnement (https://www6.rennes.inrae.fr/u3e/PRESENTATION/Organisation/Pole-MIAME). M.D. PhDs was financed by University of Pau and Pays de l'Adour (https://www.univ-pau.fr/fr/index.html) and UPV/EHU (https://www.ehu.eus/es/home). Field work used resources from the IE ECP Experimental Facility of the UMR Ecobiop (https://www6.bordeaux-aquitaine.inrae.fr/ie-ecp-ecobiop).

Additional Editor Comments:

This is an interesting paper exploring how habitat heterogeneity may promote species diversity. The focus on sea lamprey breeding sites to explore this idea is excellent. Generally, the data presented are robust. However, both reviewers point out areas which need attention to improve the overall impact of this work. I agree with their comments, especially related to explaining and justifying the experimental design (i.e., controls). Additionally, there are section which need additional details to ensure the paper is coherent across all sections.

Reviewers' comments:

Reviewer's Responses to Questions

**Comments to the Author**

1. Is the manuscript technically sound, and do the data support the conclusions?

Reviewer #1: Partly

Reviewer #2: No

2. Has the statistical analysis been performed appropriately and rigorously? 

Reviewer #1: I Don't Know

Reviewer #2: Yes

3. Have the authors made all data underlying the findings in their manuscript fully available?

Reviewer #1: Yes

Reviewer #2: No

4. Is the manuscript presented in an intelligible fashion and written in standard English?

Reviewer #1: Yes

Reviewer #2: Yes

5. Review Comments to the Author

Reviewer #1: This manuscript assesses whether nests constructed by sea lamprey in rivers alter benthic macroinvertebrate communities. This was undertaken by collecting macroinvertebrate samples from above, in the pit of, and in the mound of, 30 sea lamprey nests. Whilst this submission has interesting results and is of potential value to the scientific literature, I have some major concerns that must be addressed prior to publication.

The first of these is that I do not think that the ‘control’ zones are independent controls from the sea lamprey nests. The pits are large; 14 cm lower than the control area of the riverbed, and > 1 m in diameter. The control samples were collected only 20 – 50 cm upstream of the pit area, which is too close to be considered truly independent. For example, the large pits constructed by the lamprey will likely affect flow hydraulics (a key variable shown to be significantly related to macroinvertebrate communities in the study) in the upstream control area. Indeed, the authors allude to this in the discussion paragraph from line 299, where they say how the nests may affect the local movement of macroinvertebrates, altering the communities of gravels near the nests. Therefore, these control samples/zones are not appropriately independent of the nests to draw some of the conclusions in the discussion about the relative abundance, diversity and assemblage change driven by the nests compared to the control. Currently, it is possible to say that sea lamprey nests locally alter macroinvertebrate communities within the nest (between the upstream, pit, and mound features), but it is not possible to compare these to control conditions, as an independent control has not been collected. That said, this work is interesting and has importance in the field (even without having the true controls to compare the results to), but rewriting of parts of the manuscript and appropriately adjusting the scope of the discussion/conclusions would be needed, given that there are not true control samples present.

Secondly, the description of the statistical analysis needs to be written more clearly, with a clear signposting as to which test was used to analyse the data for each of the results subsections and questions. I found it difficult to work out which tests had been used to obtain each result, as the results section does not give the statistical results in text. Further, there are some analyses missing from this section, such as the generalized linear model to investigate if current affected the number of taxa, reported on Line 2020. Therefore, this section needs to have an improved structure, and to include all statistical analyses, and so that the statistical methods correspond with specific research questions / areas of analysis undertaken.

Lastly, the results of statistical tests need to be written in text to support assertions that are made. This is particularly needed in the ‘diversity’ section of the results.

I have also included line by line comments below, which also need addressing prior to publication.

If these concerns can be met, then this manuscript has the potential to be published in PLOS ONE, but it requires adjustment given the concerns detailed.

Abstract

‘on the Nive River’ – a brief mention of where in the world, and what size over river this is, would be beneficial here

‘traduced’ – consider a different phrasing here for clarity

When stating the reduced macroinvertebrate values (1160 to 6540…), the range only reports extreme values and is not very informative. Reporting the mean or median, with standard deviation or interquartile range here would be more appropriate

Introduction

Line 13 – What are the radius zones of – is this a survey of the total number of taxa within a given survey area, which sampled habitats of different heterogeneity? Some more information on the Beisel et al. study is needed here to fully explain this point.

Line 17 – consider different terminology to ‘sample scale’, as samples can be collected from a range of scales

Line 18 – ‘substrate appears to be…’ – what aspect of the substrate are you referring to – complexity, grain size, depth? If you are referring to a combination of all these factors, consider ‘Substrate characteristics appear to be…’

Line 23 – This is the first mention of engineer species, and a definition of ecosystem engineering (with reference to the key literature of Jones et al. 1994, and perhaps zoogeomorphology, Butler 1995) is needed before the introduction of this term

Line 36-41 – do you have any photos of lamprey nests? A figure showing these structures would be really useful here

Line 43 – what was the mechanism that attracted the Simuliidae to the area with the disturbed substrate and sea lamprey carcasses? If it was the carcasses, I’m not sure of the relevance of this to the point being made, as the physical habitat modification and presence of a carcass are very different drivers of change. Please clarify this mechanism

Methods

Line 62 – Please provide more details about the river. Please provide details of the river width, bankfull width, mean river depth, bankfull river depth, typical and bankfull discharge, and a description of the sediment characteristics (e.g. D16, D50, D84 of the sediment).

Line 70 – I do not understand your point of ‘a period short enough to limit the risk of flooding’ – please clarify what you mean by this

Line 74 – I am not convinced that the control zone is independent from the nest, as an area only 20 – 50 cm upstream of a pit is likely to have its flow hydraulics influenced by having a large pit (up to 210 cm) just 20 cm downstream of this area. Rather, these samples represent an area immediately upstream of a pit, rather than a control of the macroinvertebrate community in the absence of a pit. Indeed, you allude to this in the discussion paragraph starting on line 299 – that nests may also alter macroinvertebrate movement in the local area to the nest. To compare the differences of the nest community to a control, you would need samples from an area of the river that is not immediately next to the nest area. Thus, it is incorrect to refer to these as ‘control’ areas, and these should be referred to as ‘upstream’ areas. I understand that you cannot go back and collect more data to generate true control samples, as all samples were all collected on the same day, and so this is likely not within the scope of this study. This work is interesting and has importance without having true controls to compare your results to, but some of the discussion points and conclusions would need to be adjusted accordingly given that there are not true control samples present to compare your data to.

Line 84 – Please can you be more specific about the ‘top layer of sediment’ – can you express this as a depth (either as an estimate in mm, or as a relationship to the grainsize, e.g. to twice the depth of the average surface grain size)

Line 102 – Is this the vegan package in R? If so, please state this here

Line 102 – What is the Chao1 equation? This is less commonly known than e.g. Shannon which you give equations for – which is good – please also give an equation in the methods for the Chao1 equation

Line 111 – consider changing the word ‘evolved’ which suggests change over time, as you only collected samples on one date, and so change over time cannot be inferred

Line 132-134 – ‘To avoid a nest effect we did not compare the taxa of a pit versus the mound of the same nest and therefore we randomly selected the pit and the mound to be compared in each index value.’ Please can you clarify what you mean by ‘to avoid a nest effect’ here. As the text reads, I do not understand why you selected random pits and mounds to pair together for the LRR analysis.

Line 145 – The section of data analysis should be broken down to more specifically identify which tests were used for which results section. I found it difficult to work out which tests had been used to obtain each result, as the results section does not give the statistical results in text. Further, there are some analyses missing from this section, such as the generalized linear model to investigate if current affected the number of taxa, reported on Line 2020. Therefore, please rewrite this section to include all statistical analyses, and so that the statistical methods correspond with specific research questions / areas of analysis undertaken.

Results

Line 167 – please also add the standard deviation to the reported diameter statistics in the main text

Line 173 – Table 2 – the p values here are difficult to interpret and require transformation to gain an understandable value. Please adjust these to a value that is more easily interpreted, e.g. 0.035 for the final value. For the values < 0.001, simply reporting < 0.001 is better than the less understandable e.g. <2e-16.

Line 175-177 – you have reported the range of macroinvertebrates, but the range only reports the extreme values. A much more valuable description of the data would be the mean and standard deviation – please also include these in the reporting of the results here

Line 177 – ‘it was significantly lower in pit than in control and mound’ – please give the results of the statistical test here

Line 185 – throughout this section, statistical analyses are needed to support the observations and claims

Line 187 – ‘In spite of the slight overlap of the standard error of those estimates the species richness seems to be higher in the nest than in the control’ – please support this with a statistical test, e.g. ANOVA / t-test / mixed model.

Lines 188-191 – ‘Log response ratios of the α diversity indices (Fig 4) showed an overall trend of reduced diversity and equitability for both Shannon and Pielou indices in pit and mound, compared to the average values observed in the control zone.’ Again, please quantify this reduction, supported by an appropriate statistical test

Lines 191-193 – ‘However, only the log response ratio of the Shannon index of the pit is strictly below 0 and indicates a significantly reduced diversity in pit compared to the control zone.’ What do you mean by strictly below zero? For example, please undertake a one-sample t test to see if the log response ratio is significantly lower than zero. If you have assessed this with your mixed models, please state the results of the statistical test in this section.

Lines 194-195 – ‘However the log response ratios were highly variable and none were significantly different from 0.’ Please support this with statistical test results, e.g. a one-sample t test.

Line 199 – ‘The 199 traits studied tended to be similar in mound and control but different in pit (Table 3)’ – table three shows a significant difference in the traits between mound and control in two of the four values reported, so this is inaccurate to report.

Line 213 – ‘marginally significant’ – I think you mean marginally insignificant, as these values >0.05

Line 220 – This is the first mention of generalized linear models. Please make sure that all statistical analyses undertaken are reported in the methods section.

Discussion

Line 246 – ‘The difference between our results and theirs seems not to derive from differences in macroinvertebrate assemblages, as these were relatively similar between both studies.’ You have reported in the previous sentence that the macroinvertebrate community in Hogg et al. was dominated by Chironomidae, which was not the case in your study. This suggests relatively different assemblages, so please be more specific in this comment.

Line 247 – ‘A significantly higher number of taxa in control than in mound and pit indicates that sea lamprey nests create heterogeneity but do not increase local species diversity.’ This statement is not consistent with your results, where you state that ‘The taxa richness (Fig 2B) also did not vary significantly between control (23.5 ± 3.9 taxa) and mound (21.2 ± 4.5 taxa)’ (Line 178).

Line 251 – ‘with its unsorted grainsize’ – please detail the grain size data supporting this statement in the methods or results

Line 257 – ‘At a larger scale, occurrence of rare species and so species diversity would be higher in zones with nests than in zones without them.’ Because your controls are not entirely independent of the nest zones (see my earlier comment on this), this cannot be asserted, as you do not have data showing the macroinvertebrate assemblage in areas where nest zones are absent.

Line 276 – ‘It must be noted that sand was almost absent in the study reach apart from lamprey nests and some marginal areas’ – please add information on this sand distribution to the information of the river substrate characteristics in the methods

Line 282 – ‘A fine substrate has less shelters than a coarser one. Shelters of fine substrate are more suitable for small size range of macroinvertebrates, whereas bigger macroinvertebrates are more prone to find shelter in the coarser substrate of the mound or the control.’ The amount of shelters is dependent on the sediment size, for example, for macroinvertebrates, a fine gravel substrate has more shelters than coarse cobbles and boulders. This is an area where reporting the grain size of the substrate is needed to provide context. If you have data on the differences in grain size characteristics, please report these in the results to support this assertion

Line 297 – ‘and so, sea lamprey nests globally decrease the macroinvertebrate density.’ This is a difficult statement to support from your results, because the control samples are not independent of the pits. For example, the pits may alter flow upstream and alter macroinvertebrate movement, and thus the macroinvertebrate community in the controls. To support this claim, you would need macroinvertebrate samples from areas that are truly independent of the pit locations. It may be that the nest has increased macroinvertebrate density upstream, in the pit, and in the mound, compared to an unaltered area.

Line 318 – ‘However, macroinvertebrates are at the bottom of the food chain’ – consider rephrasing this, as many are predatory, and algae, periphyton, and vegetation is the base of the food chain for herbivorous macroinvertebrates

Line 352 – ‘In addition, negative impacts of invasive populations through population reduction of their hosts may be mitigated by a positive effect through invertebrate diversity.’ The final sentence of the discussion is problematic, and I would recommend removing this. The negative effects caused by invasive sea lamprey in e.g. the Great Lakes are substantial, and the loss of large quantities of higher taxa, such as large fish that sea lamprey predate on, is not something that can be ecologically offset by alterations to macroinvertebrate communities, which fulfil different taxonomic and functional roles in an ecosystem. The shift in the relative balance of the trophic levels is also potentially problematic, and not solved by alterations to the macroinvertebrate communities.

Reviewer #2: Dear Authors!

1. Please provide lists of species in the assamblages (probalbly as supplemental data). I found mentions of only two macroinvertebrates in the paper.

2. Please specify what you mean speaking of "local level" both in Abstract and the main body of the paper, as different researches have various understanding of it. You do not provide data on the distribution of the nests you have investigated (maybe a map or at least showing the area you have investigated). You refer the paper [5], which mentions "local" in its sources only.

3. You refer to beavers activity as ecosystems transformers but lamprey nest building process is uncomparable to dam construction. As well you highlight that the substrate disturbance (historically developed in case of lamprey spawning) may cause increased bedload transport, and refer to [13], which is on Barbel fish feeding (every day and year-around process).

4. Your choice of "control" is strange for me. You probably should better explain why you have not researched communities on left or right sides from nests.

6. PLOS authors have the option to publish the peer review history of their article (what does this mean?). If published, this will include your full peer review and any attached files.

Reviewer #1: No

Reviewer #2: No

---

## [Author Response · Author response to Decision Letter 0]

7 Nov 2022

Rebuttal letter

Dear Editor and Reviewers,

Please find in this document all your comments on the manuscript with our response. The various changes are also highlighted in the file "Manuscript_marked_up".

Yours truly,

Marius Dhamelincourt, for the authors

Editor comments

We carefully assessed and modified the manuscript accordingly.

We added the following sentence in the Methods section: “No permit was required to sample invertebrates in lamprey nests, because the sampling was performed after the larvae left the nests, thereby preventing disturbance to lamprey lifecycle.”

"Functioning was financed by Pôle Gestion des Migrateurs Amphihalins dans leur Environnement. M.D. PhDs was financed by Univ. Pau and Pays Adour and UPV/EHU. Field work used resources from the IE ECP Experimental Facility of the UMR Ecobiop [71]."

"Functioning was financed by Pôle Gestion des Migrateurs Amphihalins dans leur Environnement (https://www6.rennes.inrae.fr/u3e/PRESENTATION/Organisation/Pole-MIAME). M.D. PhDs was financed by University of Pau and Pays de l'Adour (https://www.univ-pau.fr/fr/index.html) and UPV/EHU (https://www.ehu.eus/es/home). Field work used resources from the IE ECP Experimental Facility of the UMR Ecobiop (https://www6.bordeaux-aquitaine.inrae.fr/ie-ecp-ecobiop).

We removed the funding from the acknowledgments. We do not want to modify the funding statement compared to your current version.

Reviewer #1: 

This manuscript assesses whether nests constructed by sea lamprey in rivers alter benthic macroinvertebrate communities. This was undertaken by collecting macroinvertebrate samples from above, in the pit of, and in the mound of, 30 sea lamprey nests. Whilst this submission has interesting results and is of potential value to the scientific literature, I have some major concerns that must be addressed prior to publication.

The first of these is that I do not think that the ‘control’ zones are independent controls from the sea lamprey nests. The pits are large; 14 cm lower than the control area of the riverbed, and > 1 m in diameter. The control samples were collected only 20 – 50 cm upstream of the pit area, which is too close to be considered truly independent. For example, the large pits constructed by the lamprey will likely affect flow hydraulics (a key variable shown to be significantly related to macroinvertebrate communities in the study) in the upstream control area. Indeed, the authors allude to this in the discussion paragraph from line 299, where they say how the nests may affect the local movement of macroinvertebrates, altering the communities of gravels near the nests. Therefore, these control samples/zones are not appropriately independent of the nests to draw some of the conclusions in the discussion about the relative abundance, diversity and assemblage change driven by the nests compared to the control. Currently, it is possible to say that sea lamprey nests locally alter macroinvertebrate communities within the nest (between the upstream, pit, and mound features), but it is not possible to compare these to control conditions, as an independent control has not been collected. That said, this work is interesting and has importance in the field (even without having the true controls to compare the results to), but rewriting of parts of the manuscript and appropriately adjusting the scope of the discussion/conclusions would be needed, given that there are not true control samples present.

As you suggested here and in other comments, we modified the manuscript by replacing “control” with “upstream” in text, figures and tables. We justified the choice of this upstream area in methods: “[upstream] being both close to the nest and less likely to be disturbed by digging than the left, right or immediate downstream of the nest”. Furthermore, we suggested this issue in the discussion: “To complete this finding, samples collected on a substrate further from the nests than our upstream samples may help to verify than they were not negatively affected by the downstream digging activity of lampreys”.

Secondly, the description of the statistical analysis needs to be written more clearly, with a clear signposting as to which test was used to analyse the data for each of the results subsections and questions. I found it difficult to work out which tests had been used to obtain each result, as the results section does not give the statistical results in text. Further, there are some analyses missing from this section, such as the generalized linear model to investigate if current affected the number of taxa, reported on Line 2020. Therefore, this section needs to have an improved structure, and to include all statistical analyses, and so that the statistical methods correspond with specific research questions / areas of analysis undertaken. Lastly, the results of statistical tests need to be written in text to support assertions that are made. This is particularly needed in the ‘diversity’ section of the results.

All statistical results are now written in text and clearly indicated in methods, with the question they aim to answer. In the “Data analysis” part, each paragraph is related to an analysis (in the order: existence of general differences for depth, current, density and diversity as well as alpha indices considering the three zones; then, pairwise difference of density and diversity between zones and relationship with current and depth; significance of Chao1 diversity index between nest (pit+mound) and upstream; differences of traits between each zone; relationship between traits and physical characteristics). As it was less clear with a single paragraph, we divided this part into small paragraphs to clearly distinguish each part of the analysis.

I have also included line by line comments below, which also need addressing prior to publication.

If these concerns can be met, then this manuscript has the potential to be published in PLOS ONE, but it requires adjustment given the concerns detailed.

Abstract

‘on the Nive River’ – a brief mention of where in the world, and what size over river this is, would be beneficial here

We added details about the location: “a river of the south western France with a length of 79,3 km and tributary of the Adour River”.

‘traduced’ – consider a different phrasing here for clarity

We changed this phrasing by “accompanied”.

When stating the reduced macroinvertebrate values (1160 to 6540…), the range only reports extreme values and is not very informative. Reporting the mean or median, with standard deviation or interquartile range here would be more appropriate

We reported the mean and the standard deviation instead of the extreme values.

Introduction

Line 13 – What are the radius zones of – is this a survey of the total number of taxa within a given survey area, which sampled habitats of different heterogeneity? Some more information on the Beisel et al. study is needed here to fully explain this point.

We gave precisions about this study as requested: “the number of macroinvertebrate taxa in a given area (considering 0.5 to 4 m radius zones of substrate in a fourth-order stream) was higher in a heterogeneous environment with more substrate types and an elevated patchiness”.

Line 17 – consider different terminology to ‘sample scale’, as samples can be collected from a range of scales

We rephrase the sentence considered: “At a very fine scale, Boyero [9] found a positive relationship between substrate heterogeneity and taxonomic diversity across 225 cm2 samples collected within the same square meter of riffle.“ which gives more precision about the scale considered.

Line 18 – ‘substrate appears to be…’ – what aspect of the substrate are you referring to – complexity, grain size, depth? If you are referring to a combination of all these factors, consider ‘Substrate characteristics appear to be…’

Here we are referring to the substrate complexity, so we added “complexity” after “substrate”.

Line 23 – This is the first mention of engineer species, and a definition of ecosystem engineering (with reference to the key literature of Jones et al. 1994, and perhaps zoogeomorphology, Butler 1995) is needed before the introduction of this term

We provided details and added these two references: “Ecosystem engineering [12, 13] is defined as the change of availability of resources to other species due to physical changes in biotic or abiotic materials, directly or indirectly mediated by organisms. During the process they modify the habitats or create new ones. Effects of engineer species [14], alongside with the interaction between hydraulics and the substrate, are abiotic and biotic processes influencing heterogeneity. Together, they affect habitat heterogeneity, which in turn can affect ecosystem functioning [6].”

Line 36-41 – do you have any photos of lamprey nests? A figure showing these structures would be really useful here

Yes, we added a figure as requested.

Line 43 – what was the mechanism that attracted the Simuliidae to the area with the disturbed substrate and sea lamprey carcasses? If it was the carcasses, I’m not sure of the relevance of this to the point being made, as the physical habitat modification and presence of a carcass are very different drivers of change. Please clarify this mechanism

We added details: “possibly resulting from the association of the cleared substrate with suitable characteristics and food supply from dead lampreys” which refer to the hypothesis given in the article, a combination of the effects of the substrate + the carcasses, not the carcasses only.

Methods

Line 62 – Please provide more details about the river. Please provide details of the river width, bankfull width, mean river depth, bankfull river depth, typical and bankfull discharge, and a description of the sediment characteristics (e.g. D16, D50, D84 of the sediment).

We added some details as requested: river width: 18 m; bankfull width: 25 m; mean river depth: 52 cm; average discharge: 21 m3/s (from hydro.eaufrance.fr). However, we do not have precise information on sediment characteristics, bankfull discharge and bankfull river depth.

Line 70 – I do not understand your point of ‘a period short enough to limit the risk of flooding’ – please clarify what you mean by this

We modified the sentences considered: “To avoid disturbing sea lamprey spawning, macroinvertebrate samples were collected on July 7 2020, at the end of the reproductive season, after two weeks of not detecting any lamprey near the nests. We considered these two weeks a period sufficient to allow for stability of local invertebrate assemblages [27], whereas reducing the risk of scouring by floods, which are frequent at the study site during late spring and early summer.”

Line 74 – I am not convinced that the control zone is independent from the nest, as an area only 20 – 50 cm upstream of a pit is likely to have its flow hydraulics influenced by having a large pit (up to 210 cm) just 20 cm downstream of this area. Rather, these samples represent an area immediately upstream of a pit, rather than a control of the macroinvertebrate community in the absence of a pit. Indeed, you allude to this in the discussion paragraph starting on line 299 – that nests may also alter macroinvertebrate movement in the local area to the nest. To compare the differences of the nest community to a control, you would need samples from an area of the river that is not immediately next to the nest area. Thus, it is incorrect to refer to these as ‘control’ areas, and these should be referred to as ‘upstream’ areas. I understand that you cannot go back and collect more data to generate true control samples, as all samples were all collected on the same day, and so this is likely not within the scope of this study. This work is interesting and has importance without having true controls to compare your results to, but some of the discussion points and conclusions would need to be adjusted accordingly given that there are not true control samples present to compare your data to.

We changed “control zone” into “upstream zone” in all the manuscript, figures and tables. We also modified the discussion based on this comment.

Line 84 – Please can you be more specific about the ‘top layer of sediment’ – can you express this as a depth (either as an estimate in mm, or as a relationship to the grainsize, e.g. to twice the depth of the average surface grain size)

We indicated that the stirring did not exceed 50 mm depth, and also specified that all the area covered by the nest frame was stirred.

Line 102 – Is this the vegan package in R? If so, please state this here

Indeed, we added this information as requested, for the Shannon index: “calculated using the diversity function of the vegan package [27, version 2.5-7]”. We also modified the location of the sentence indicating that all analyses were made using R: “All analyzes were performed using R software version 4.1.0 and a significance level of 0.05”.

Line 102 – What is the Chao1 equation? This is less commonly known than e.g. Shannon which you give equations for – which is good – please also give an equation in the methods for the Chao1 equation

We are now giving the equation as requested.

Line 111 – consider changing the word ‘evolved’ which suggests change over time, as you only collected samples on one date, and so change over time cannot be inferred

We changed the word “evolved” into “differed”.

Line 132-134 – ‘To avoid a nest effect we did not compare the taxa of a pit versus the mound of the same nest and therefore we randomly selected the pit and the mound to be compared in each index value.’ Please can you clarify what you mean by ‘to avoid a nest effect’ here. As the text reads, I do not understand why you selected random pits and mounds to pair together for the LRR analysis.

We added details about this choice right after the sentence you quoted: “Indeed, pit and mound of the same nest may not be independent, and this analysis aimed at describing the heterogeneity of communities between pits and mounds in general, without the influence of similarities existing between adjacent samples.”

Line 145 – The section of data analysis should be broken down to more specifically identify which tests were used for which results section. I found it difficult to work out which tests had been used to obtain each result, as the results section does not give the statistical results in text. Further, there are some analyses missing from this section, such as the generalized linear model to investigate if current affected the number of taxa, reported on Line 2020. Therefore, please rewrite this section to include all statistical analyses, and so that the statistical methods correspond with specific research questions / areas of analysis undertaken.

We added details about the two-sample t-test used for Chao1 diversity index: “A two-sample t-test was used to test the significance of Chao1 diversity index calculated for nest (pit + mound) and upstream zone”. We detail the general linear model analysis in the first paragraph of the section instead of the last one. To follow the same logic, we moved the results of this analysis from the “redundancy analysis” part to the “macroinvertebrate density and diversity” part (earlier in the results). Following your next comments, we added to every statement based on a statistical analysis the test results in the text.

Results

Line 167 – please also add the standard deviation to the reported diameter statistics in the main text

We recalled this information, as stated in Table 1.

Line 173 – Table 2 – the p values here are difficult to interpret and require transformation to gain an understandable value. Please adjust these to a value that is more easily interpreted, e.g. 0.035 for the final value. For the values < 0.001, simply reporting < 0.001 is better than the less understandable e.g. <2e-16.

We modified the p-values of the Table 2 accordingly.

Line 175-177 – you have reported the range of macroinvertebrates, but the range only reports the extreme values. A much more valuable description of the data would be the mean and standard deviation – please also include these in the reporting of the results here

As done in the abstract, we replaced the extreme values with the mean ± standard deviation.

Line 177 – ‘it was significantly lower in pit than in control and mound’ – please give the results of the statistical test here

We added the results of the Tukey’s tests performed for abundance results: “It was significantly lower in pit than in upstream and mound (Tukey test: Df = 58, t.ratio = 3.529, p-value = 0.0023 and Df = 58, t.ratio = -3.705, p-value = 0.0014 respectively) but there were no differences between control and mound (Tukey test: Df = 58, t.ratio = -0.175, p-value = 0.9832)”.

Line 185 – throughout this section, statistical analyses are needed to support the observations and claims

We considered your concerns in the following points. We also added the statistics of Tukey’s tests for the analysis on number of taxa: “The taxa richness (Fig 3B) also did not vary significantly between upstream (23.5 ± 3.9 taxa) and mound (21.2 ± 4.5 taxa; Tukey test: Df = 58, t.ratio = 2.293, p-value = 0.0647) but was higher in these two zones than in pit (18.6 ± 3.9 taxa; Tukey test: Df = 58, t.ratio = 5.066, p-value = ≤ 0.0001 for pit compared to upstream and Tukey test: Df = 58, t.ratio = -2.772, p-value = 0.0201 for pit compared to mound).”

Line 187 – ‘In spite of the slight overlap of the standard error of those estimates the species richness seems to be higher in the nest than in the control’ – please support this with a statistical test, e.g. ANOVA / t-test / mixed model.

We indicated the results of a two-sample t test: “In spite of the slight overlap of the standard error of those estimates the species richness was higher in the nest than in upstream, as supported by results of two-sample t-test (Df = 44.739, t = 4.5017, p-value ≤ 0.0001)”.

Lines 188-191 – ‘Log response ratios of the α diversity indices (Fig 4) showed an overall trend of reduced diversity and equitability for both Shannon and Pielou indices in pit and mound, compared to the average values observed in the control zone.’ Again, please quantify this reduction, supported by an appropriate statistical test

Log response ratios do not need to be accompanied by a statistical test. When this ratio does not overlap 0, the effect is considered either negative (ratio under 0) or positive (ratio above 0). That is why we added a horizontal red line at 0 value, which indicates this significance (see DOI:10.1038/s41598-020-60454-z for example). However, we assume that this approach needs more details in the methods and in the figure legends. So, we added “When the LRR standard deviation does not cross the value "0", it means that the effect is significantly positive (above 0) or negative (below 0; [34]).” We added the precise value of the reduction considered for Shannon in pit compared to the upstream zone: “(- 10.5 ± 10.2 %)”, as it was the only LRR significant.

Lines 191-193 – ‘However, only the log response ratio of the Shannon index of the pit is strictly below 0 and indicates a significantly reduced diversity in pit compared to the control zone.’ What do you mean by strictly below zero? For example, please undertake a one-sample t test to see if the log response ratio is significantly lower than zero. If you have assessed this with your mixed models, please state the results of the statistical test in this section.

Here, the explanation is the same as for your precedent concern. Log response ratios are significant when the standard deviation does not cross 0. Consequently, no other test is needed.

Lines 194-195 – ‘However the log response ratios were highly variable and none were significantly different from 0.’ Please support this with statistical test results, e.g. a one-sample t test.

Same answer than the two previous points.

Line 199 – ‘The 199 traits studied tended to be similar in mound and control but different in pit (Table 3)’ – table three shows a significant difference in the traits between mound and control in two of the four values reported, so this is inaccurate to report.

We have changed the phrasing to make this result more nuanced: “The traits studied tended to be more similar in mound and upstream than between these two locations and the pit”.

Line 213 – ‘marginally significant’ – I think you mean marginally insignificant, as these values >0.05

Indeed, we modified this word.

Line 220 – This is the first mention of generalized linear models. Please make sure that all statistical analyses undertaken are reported in the methods section.

We described the analysis before: “To test the effect of current and depth on abundance and taxa diversity, a general linear model was used with current and depth as covariates and following a Gaussian distribution.” However, we described a “general linear model” instead of a “generalized linear model”, the latter being a mistake now addressed: “General Linear Models indicated that…”.

Discussion

Line 246 – ‘The difference between our results and theirs seems not to derive from differences in macroinvertebrate assemblages, as these were relatively similar between both studies.’ You have reported in the previous sentence that the macroinvertebrate community in Hogg et al. was dominated by Chironomidae, which was not the case in your study. This suggests relatively different assemblages, so please be more specific in this comment.

The difference in communities here discussed is the taxa occurrence, not their abundance. But we agree it was unclear and we rephrased the sentence: “The difference between our results and theirs seems not to derive from differences in macroinvertebrate taxa composition, as it was relatively similar between both studies (with Chironomidae, Hydropsychidae, Heptageniidae and Ephemerellidae dominant in [24] and abundant in the Nive River: Fig 12 and Table 4)”.

Line 247 – ‘A significantly higher number of taxa in control than in mound and pit indicates that sea lamprey nests create heterogeneity but do not increase local species diversity.’ This statement is not consistent with your results, where you state that ‘The taxa richness (Fig 2B) also did not vary significantly between control (23.5 ± 3.9 taxa) and mound (21.2 ± 4.5 taxa)’ (Line 178).

Indeed, this sentence was not consistent with the results.we replaced it to discuss the results of the Chao index: “The significantly higher number of taxa in nest (mound and pit combined) than upstream, highlighted by the Chao1 index, indicates that sea lamprey nests create heterogeneity but also increase local species diversity.”

Line 251 – ‘with its unsorted grainsize’ – please detail the grain size data supporting this statement in the methods or results

We do not have precise data on grain size. However, this statement was added to highlight that the upstream zone is not sorted by lampreys compared to the nest, where lampreys remove coarsest elements from the pit to build the mound. To make it clearer, we added this sentence: “(due to the absence of sorting by lamprey during the nest building compared to the nest)”.

Line 257 – ‘At a larger scale, occurrence of rare species and so species diversity would be higher in zones with nests than in zones without them.’ Because your controls are not entirely independent of the nest zones (see my earlier comment on this), this cannot be asserted, as you do not have data showing the macroinvertebrate assemblage in areas where nest zones are absent.

We removed this sentence.

Line 276 – ‘It must be noted that sand was almost absent in the study reach apart from lamprey nests and some marginal areas’ – please add information on this sand distribution to the information of the river substrate characteristics in the methods

As previously stated, we do not have precise information on grain size distribution. However, we agree that this sentence suggests that a granulometry analysis was made. So, we added some details: “It must be noted that we visually noticed an absence of sand in the substrate surface of the study reach”.

Line 282 – ‘A fine substrate has less shelters than a coarser one. Shelters of fine substrate are more suitable for small size range of macroinvertebrates, whereas bigger macroinvertebrates are more prone to find shelter in the coarser substrate of the mound or the control.’ The amount of shelters is dependent on the sediment size, for example, for macroinvertebrates, a fine gravel substrate has more shelters than coarse cobbles and boulders. This is an area where reporting the grain size of the substrate is needed to provide context. If you have data on the differences in grain size characteristics, please report these in the results to support this assertion

We do not have this data. As already done for other comments, we nuanced this hypothesis by adding: “Data on the grain size distribution of nest zones may help to confirm this hypothesis”.

Line 297 – ‘and so, sea lamprey nests globally decrease the macroinvertebrate density.’ This is a difficult statement to support from your results, because the control samples are not independent of the pits. For example, the pits may alter flow upstream and alter macroinvertebrate movement, and thus the macroinvertebrate community in the controls. To support this claim, you would need macroinvertebrate samples from areas that are truly independent of the pit locations. It may be that the nest has increased macroinvertebrate density upstream, in the pit, and in the mound, compared to an unaltered area.

Your concern about the controls is understandable. We added it in the discussion, right after the sentences you quoted: “To complete this finding, samples collected on a substrate further from the nests than our upstream samples may help to verify than they were not negatively affected by the downstream digging activity of lampreys”.

Line 318 – ‘However, macroinvertebrates are at the bottom of the food chain’ – consider rephrasing this, as many are predatory, and algae, periphyton, and vegetation is the base of the food chain for herbivorous macroinvertebrates

We rephrased it: “However, macroinvertebrates have a basal to intermediate position on the food chain”.

Line 352 – ‘In addition, negative impacts of invasive populations through population reduction of their hosts may be mitigated by a positive effect through invertebrate diversity.’ The final sentence of the discussion is problematic, and I would recommend removing this. The negative effects caused by invasive sea lamprey in e.g. the Great Lakes are substantial, and the loss of large quantities of higher taxa, such as large fish that sea lamprey predate on, is not something that can be ecologically offset by alterations to macroinvertebrate communities, which fulfil different taxonomic and functional roles in an ecosystem. The shift in the relative balance of the trophic levels is also potentially problematic, and not solved by alterations to the macroinvertebrate communities.

We removed this sentence as suggested.

Reviewer #2: 

Dear Authors!

1. Please provide lists of species in the assamblages (probalbly as supplemental data). I found mentions of only two macroinvertebrates in the paper.

Now a complete list of taxa for each nest and each zone is provided as supplementary material.

2. Please specify what you mean speaking of "local level" both in Abstract and the main body of the paper, as different researches have various understanding of it. You do not provide data on the distribution of the nests you have investigated (maybe a map or at least showing the area you have investigated). You refer the paper [5], which mentions "local" in its sources only.

We added details on the local-scale described: “(i.e. nest and adjacent substrate)”. As requested we added a map showing the location of the site and the distribution of the nests (Fig 2). For the reference [5], indeed the paper did not develop “local” level but it is quoted to highlight the variability of effects across scales in general, not solely at the local level.

3. You refer to beavers activity as ecosystems transformers but lamprey nest building process is uncomparable to dam construction. As well you highlight that the substrate disturbance (historically developed in case of lamprey spawning) may cause increased bedload transport, and refer to [13], which is on Barbel fish feeding (every day and year-around process).

Indeed, we refer to other species with different mechanisms than sea lamprey nest-building, but this was done on purpose to depict the complexity and richness of ecosystem engineering in rivers. The next paragraph specifically focuses on the nest-building process by fish in general and lamprey in particular.

4. Your choice of "control" is strange for me. You probably should better explain why you have not researched communities on left or right sides from nests.

We added details to justify this choice: “being both close to the nest and less likely to be disturbed by digging than the left, right or immediate downstream of the nest”.

---

## [Editor Report · Decision Letter 1]

16 Nov 2022

Sea lamprey nests promote the diversity of benthic macroinvertebrate assemblages

PONE-D-22-24495R1

Dear Dr. Dhamelincourt,

We’re pleased to inform you that your manuscript has been judged scientifically suitable for publication and will be formally accepted for publication once it meets all outstanding technical requirements.

Kind regards,

Michael A Chadwick, PhD

Academic Editor

PLOS ONE

Additional Editor Comments (optional):

Thank you for this resubmission. You have thoughtfully consider and addressed all of the comments provided by the 2 reviewers. The changes made have improved the work. The replacement of the word "control" works well and the inclusion of the supplementary materials will be of use to readers.
---

## [Editor Report · Acceptance letter]

17 Nov 2022

PONE-D-22-24495R1 

Sea lamprey nests promote the diversity of benthic
macroinvertebrate assemblages 

Dear Dr. Dhamelincourt:

I'm pleased to inform you that your manuscript has been deemed suitable for publication in PLOS ONE. Congratulations! Your manuscript is now with our production department. 

Kind regards, 

on behalf of

Dr. Michael A Chadwick 

Academic Editor

PLOS ONE